# Changes in Erythrocyte Omega-3 Fatty Acids in German Employees upon Dietary Advice by Corporate Health

**DOI:** 10.3390/nu12113267

**Published:** 2020-10-25

**Authors:** Dietrich Rein, Matthias Claus, Wolfgang Frosch, Winfried März, Stefan Lorkowski, Stefan Webendoerfer, Thorsten Schreiner

**Affiliations:** 1BASF SE, Nutrition and Health, Human Nutrition, 68623 Lampertheim, Germany; 2BASF SE, Corporate Health Management, 67056 Ludwigshafen, Germany; matthias.claus@basf.com (M.C.); wolfgang.frosch@basf.com (W.F.); stefan.webendoerfer@basf.com (S.W.); thorsten.schreiner@basf.com (T.S.); 3Synlab Academy, Synlab Holding Germany GmbH, P5, 7, 68167 Mannheim, Germany; Winfried.Maerz@synlab.com; 4Clinical Institut for Medical and Chemical Laboratory Diagnostics, Medical University of Graz, Auenbrugger Platz, 8036 Graz, Austria; 5Medical Clinic V, Medical Faculty Mannheim, University of Heidelberg, Theodor-Kutzer-Ufer 1-3, 68167 Mannheim, Germany; 6Competence Cluster for Nutrition and Cardiovascular Health (nutriCARD) Halle-Jena-Leipzig, 07743 Jena, Germany; stefan.lorkowski@uni-jena.de; 7Institute of Nutritional Sciences, Friedrich Schiller University Jena, 07743 Jena, Germany

**Keywords:** eicosapentaenoic acid, docosahexaenoic acid, employee health state, omega-3 index

## Abstract

Background: The erythrocyte ratio of eicosapentaenoic acid (EPA) and docosahexaenoic acid (DHA) over total fatty acids, the omega-3 index (O3I), has been suggested as an overall health marker and to motivate corporate health recommendations. We set out to assess the O3I status in a working population, the differences between normal and rotating shift employees, the consumption of omega-3 rich food and whether recommendations to increase intake of omega-3 rich foods can improve the O3I. Methods: Employees registered for their occupational health check-up were offered to participate in a pre-post study at the Ludwigshafen (Germany) site including an assessment of their O3I at baseline and after 4 months (follow-up) and two subsequent food frequency questionnaires. For those with O3I below 8%, a recommendation was provided to increase the intake of omega-3 fatty acid rich food and to take advantage of the employees’ catering service with its enhanced fatty seafood offer during the study period. Dietary intake of EPA and DHA, erythrocyte fatty acid profiles, clinical and lifestyle parameters were assessed. Results: In 500 employees (26.6% female, 21–64 years, median age: 47 years [IQR: 37–53]), at baseline the overall mean O3I was 4.1 ± 1.1% (99.6% of O3I assessed were below 8%), higher in women, in participants with “normal” body weight, upper employment grade, and non-smokers, but not different between regular and rotating shift workers. The three fifths of the cohort also participating in the follow-up increased their EPA and DHA intake by 0.1 g/d and their O3I by 0.5 percentage points. Conclusion: This study provides essential data on omega-3 erythrocyte concentrations in a clinically healthy German working population and the challenges of increasing the O3I with dietary recommendations even in study participants motivated to follow up on their omega-3 status.

## 1. Introduction

The polyunsaturated omega-3 fatty acids eicosapentaenoic acid (EPA) and docosahexaenoic acid (DHA) have important physiological functions in the human body. They are vital components of all cell membranes [1,2], and are precursors of numerous signaling molecules [3,4], although the differences in the physiological functions and dietary importance of EPA vs. DHA are still insufficiently understood [5]. These omega-3 fatty acids are crucial to all phases of human development from pregnancy to aging [6,7]. In the adult organism, highest concentrations of EPA and DHA are found in brain, nerve cells and the retina of the eye [2].

EPA and DHA are conditionally indispensable as the human body can synthesize them from essential α-linolenic acid. However, the conversion of α-linolenic acid to EPA and to DHA is estimated to be low, at least for men (DHA about 1%) [8], and is somewhat higher for women (DHA 9–22%) [9]. Moreover, genetic differences and conditions such as pregnancy affect the efficacy of this conversion [10]. Intake recommendations for EPA and DHA vary widely, but range mostly between 0.2 and 0.5 g/d for healthy adults. For this study, we recommended the intake of at least 0.5 g EPA and DHA per day based on international expert opinions [11,12].

Omega-3 blood concentrations can be assessed in serum or plasma triglycerides and phospholipids, but the currently most widely accepted specimen reflecting EPA and DHA status in humans is red blood cell membrane lipids. Erythrocyte lipids are independent of acute food intake and are thought to reflect whole body tissue status and longer-term food intake. The O3I is calculated as the ratio of EPA plus DHA over the total quantified fatty acids [13,14]. Typical O3I values for adults increase with age varying between 3% and 5% and most studies show higher O3I for women [15,16]. Dietary omega-3 fatty acids appear to be efficiently incorporated into blood compartments [17,18] and monitoring the O3I indicates the efficacy of food- or supplement-derived EPA and DHA incorporation [19]. For our study, we set the goal of an O3I above 8% based on previous recommends supported by cardiovascular risk reduction [13,20].

There is evidence that the dietary and pharmacological supply of EPA and DHA may provide health benefits [21,22]. Indeed, increasing dietary EPA and DHA intake through foods or supplements has been associated with the improvement of cardiovascular risk factors, such as blood lipids, blood pressure, heart rate and heart rhythm, platelet aggregation, endothelial function and inflammation [23]. EPA- and DHA-containing supplements have been associated with lowering cardiovascular risk in a concentration-dependent manner [24,25]. Still, although the intake of EPA and DHA reduces plasma triglycerides and may be slightly protective of certain heart and circulatory disease conditions, the overall evidence for the effects from consuming EPA and DHA from fish or supplements is debated [21].

Fatty fish, seafood, krill oil and certain algae oils are the main dietary sources for EPA and DHA [26]. Recently, the successful incorporation of EPA and DHA from a transgenic plant oil crop into human blood lipids was demonstrated in a clinical trial [27]. However, global intake of EPA and DHA and overall values for the O3I vary significantly [16]. Previously measured omega-3 fatty acid concentrations in a German cardiovascular disease risk population ranged between 5% and 6% [28].

Occupational health campaigns in company settings intend to identify health concerns in the working population and to motivate long-term prevention strategies [29]. They can also identify employees’ work-related lifestyle imbalances enabling, for example, improvements in back health [30], or sleep patterns of shift workers [31]. Ideally, company health services identify the most effective health topics to provide benefit to both, employee and employer. For this study, our health department of a large employer investigated cardiovascular wellbeing focusing on omega-3 fatty acid status and intake. The results provide orientation for occupational health strategies towards improved employee health and health care cost savings.

Specifically, we tested the hypotheses that in a southern German cohort of employees, (i) the average dietary intake of EPA and DHA is below 0.5 g, and the average O3I is below 8%, (ii) the employees on rotating shifts do not differ with respect to their O3I from day shift employees, and (iii) the O3I can be increased through dietary recommendations and an improved offer of fatty seafood in the employee’s canteen. 

## 2. Materials and Methods 

### 2.1. Study Setting and Participants

For this monocentric baseline and follow-up survey without a control group, participants were enrolled between October and December 2018 (Baseline) and followed-up between March and April 2019 (Follow-Up; Figure 1). Participants were invited to participate in the “Omega-3 Employee Study” by the corporate health management within the voluntary occupational health check-up that is offered to employees once every three years and includes anamnesis, medical consultation, clinical examination, laboratory diagnostic, urine diagnostic, fecal occult blood test and a questionnaire for self-examination of the personal health status. Participation in this study was completely voluntary, included an additional blood draw, assessment of food intake using omega-3 fatty acid intake specific questionnaires and a lifestyle assessment at baseline and follow-up. There were no specific study inclusion or exclusion criteria except for the wish to participate. Due to the predominantly exploratory pilot character of our study, we did not carry out a formal power calculation for our primary outcome variable. Each participant was attended by his or her assigned occupational health physician, signed a written study consent, and received a study information. The study was approved by the ethics commission of the Landesärztekammer Rheinland-Pfalz (application number 2018-13589). 

Participants filled out the omega-3 intake-focused dietary intake questionnaire, participated in an additional physical examination, and a blood draw for a complete fatty acid profile determination, a lipoprotein (a) (Lp(a)) measurement, a medical anamnesis and a personal consultation with the physician. Participants were informed within two weeks after their baseline examination about their usual health check-up parameters and about their O3I values. Moreover, every participant obtained a short personal consultation about the health benefits of higher O3I values for cardiovascular health and about the cut-offs discussed in the scientific community [13]. A handout was made available to study participants for information and reference. Participants were recommended to take advantage of the improved offer of fatty seafood in the canteen consisting of two additional fatty fish meals per week. For the follow-up sampling, volunteers were invited by e-mail for a second visit to the health center approximately 120 days after their baseline assessment.

### 2.2. Occupational Health Check-Up

The baseline occupational health check-up consisted of a written questionnaire, physical examination, blood sampling and anamnesis by the occupational health physician (blood pressure, pulse, body weight, height, and abdominal circumference). Laboratory diagnostics (carried out by the clinic of the city of Ludwigshafen am Rhein) included a differential blood cell count, liver enzymes (aspartate aminotransferase (ASAT), alanine aminotransferase (ALAT) and γ-glutamyltransferase (GGT)), uric acid, creatinine, total cholesterol, triglycerides, the HDL/LDL cholesterol ratio, serum glucose, glycated hemoglobin (HbA1c), C-reactive protein (CRP) and basal thyroid-stimulating hormone (TSH). The questionnaire included modules on regeneration and stress management and back health. All participants subsequently received a doctor’s letter with a summary assessment and individual recommendations.

### 2.3. Omega-3 Fatty acid Intake Specific Questionnaire

The questionnaire for baseline and follow-up were largely identical apart from an additional module examining eating habits during follow-up. Baseline questions related to food intake in the previous 4 weeks, follow-up questions related to food intake between baseline and follow-up. The focus of the questions was on the intake of omega-3 fatty acids, i.e., consumption of fish, seafood, omega-3 fortified food and dietary supplements. In addition, the questionnaire interrogated the frequency of visiting and satisfaction with the company restaurants. Most of the items were adapted from the validated nutritional questionnaire from the DEGS1 study (Study on Adult Health in Germany, Robert Koch Institute, Berlin, Germany) [32]. Several questions on omega-3 supplementation, time fasted before participation, and visiting company canteens were included in addition. 

### 2.4. Analysis of Blood Parameters

Erythrocyte fatty acid composition was analyzed as described previously [13]. In brief, fatty acid methyl esters were generated from erythrocytes by acid transesterification and analyzed by gas chromatography using a GC2010 gas chromatograph (Shimadzu, Duisburg, Germany) equipped with a 100-m SP2560 column (Supelco, Bellefonte, PA, USA) and using hydrogen as carrier gas. Fatty acids were identified by comparison with a standard mixture of fatty acids characteristic of erythrocytes. The results are given as a percentage of total identified fatty acids after response factor correction.

Lipoprotein (a) was quantified using a Lp(a) specific antibody-based turbidity method with absorption at 694 nm on an Atellica™ CH System (Siemens Healthcare, Erlangen, Germany). Limit of detection (LOD) for Lp(a) was 6 mg/dL. Of the 500 participants at baseline, 145 (29%) had Lp(a) values below LOD.

### 2.5. Variable Definitions

The intake of EPA and DHA was derived from the consumption of cold and warm fish recorded in the questionnaire. Fish typically containing <4% fat was classified as lean (e.g., pollack, pikeperch, cod, redfish, hake, plaice, sea bass, trout, char), whereas fish containing ≥4% fat as fatty (e.g., salmon, mackerel, sardine, tuna, herring, maatjes) [33]. The average EPA and DHA content for low-fat and fatty fish species was estimated from the US Nutrient Content database [34], assuming that 1 g lean fish contributes 3 mg, and 1 g of fatty fish contributes 15 mg of dietary EPA plus DHA. The O3I was categorized with respect to their suggested level of cardioprotection (CP) as <4% “least CP”, 4–8% “average CP” and >8% “largest CP” [13]. Lipoprotein (a) values were classified into three categories, “no indication of Lp(a) associated atherogenic risk” (≤30 mg/dL), “Lp(a) associated risk slightly increased” (30–60 mg/dL) and “Lp(a) associated risk significantly increased” (>60 mg/dL).

Age at the time of examination (categorized into groups: <35, 35–39, …, ≥55), gender, occupational group (manual workers, skilled/supervisory workers, managerial staff), and working time system (day work, shift work) were directly extracted from employees’ records. The classification of workers into occupational groups was assumed to roughly represent the socioeconomic status of the participants. Whereas manual workers work in the production lines and have a physically demanding job, skilled/supervisory workers usually have a better education and commonly perform commercial activities or office work, while managerial staff hold an academic degree. Regarding shift work, a 4 × 12 h-rotating shift schedule is predominantly used within the company (>90% of all shift workers at the Ludwigshafen site in 2019), with shifts starting at 6 a.m. and ending at 6 p.m. with a subsequent leisure time of 24 h and the following shift starting in the evening of the subsequent day at 6 p.m. and ending at 6 a.m. with a subsequent leisure time of 48 h. Smoking status (smoker, former smoker, non-smoker), height and weight, and waist circumference (in cm) were obtained during medical anamnesis by the responsible occupational health physician. Body mass index (BMI) was calculated using the formula weight [kg]/height squared [m^2^] and has been categorized according to the World Health Organization (normal weight: BMI < 25 kg/m^2^, overweight: 25 kg/m^2^ ≤ BMI < 30 kg/m^2^, obesity: ≥30 kg/m^2^).

From the study-specific questionnaire, we extracted data on the duration of fasting before blood was drawn (0–1, 1–2, 2–3, 3–4, ≥5 h [=fasted]), vegetarianism (strict/predominantly/no) and/or veganism (yes/no), frequency and amount of consumed warm (fatty/lean) and cold fish, duration, frequency and amount of recent omega-3 fatty acid supplementation, frequency of visiting canteens within the company (never, 1 time per month, 2–3 times per month …, daily (Monday to Friday)), and frequency of consumption of fatty fish offered by the canteens (never, 1 time per 4-weeks, 2–3 times per month, 1 time per week, several times per week, daily (Monday to Friday)). Based on information regarding the consumption of fish, we set up several scores on the average consumption of the different types of fish (in g/day) by multiplying the frequency of the consumption of fish (never, 1 time in the last 4 weeks, 2–3 times in the last 4 weeks, 1–2 times per week, 3–4 times per week, ≥5 times per week) by its corresponding amount (≤1/4, 1/2, 1, 2, ≥3 portions). One portion of fish was defined as 90 g, such that a person consuming, e.g., 1/2 portion of warm fatty fish (=45 g) 2–3 times in the last 4 weeks received a score of 4.0 g/d (= [2.5 times ∗ 45 g]/28 days).

### 2.6. Statistical Assessment

The results were presented descriptively for continuous variables with arithmetic mean and standard deviation (SD) or median and interquartile range (IQR) for normally and non-normally distributed variables, respectively, for categorical variables with absolute and relative frequencies. Normality of data was examined using the Shapiro–Wilk test. For the O3I, we noted a slight deviation from normality (right-tailed) within our sample. For ease of comparison with earlier studies on the topic, we decided to provide arithmetic means (SD) in addition to the median (IQR) for the O3I. Mann–Whitney U tests or Kruskal–Wallis tests were used to test for statistically significant differences in the (continuous) O3I between two or among more than two groups. Pearson’s chi-square test or Fisher’s exact test was used to compare multiple categorical variables. In order to test for significant differences in O3I levels between shift and day workers at baseline, we used a Mann–Whitney U test for the bivariate case and we applied a multiple linear regression model adjusted for age, occupational status, smoking, BMI, vegetarian diet, and eating in company restaurants which were considered major potential confounders for the association between working time system and O3I. We used linear mixed models to estimate within and between-group changes in O3I between baseline and follow-up for the main sociodemographic and occupational-related variables separately. In all, we included the categorical group variable (e.g., categorical age, gender, BMI, etc.), time (baseline/follow-up) and group–time interaction terms as fixed effects and a participant-specific random-intercept with an unstructured correlation structure for all models. *p*-values < 0.05 were considered statistically significant. Statistical assessments were performed using STATA version 15.1 (Stata Corp, College Station, TX, USA).

## 3. Results

### 3.1. Baseline

#### 3.1.1. Study Participants

Of the 509 enrolled participants, 500 were included in the analysis; five were excluded because of missing laboratory data and 4 because of not completing their questionnaire (Figure 1). Baseline characteristics in our sample and compared to the overall company population on 1st January 2019 are provided in Table 1. Participants were between 21 and 64 years old (median: 47 years [IQR: 37–53]) and the overall median BMI was 25.7 [IQR: 23.5–28.5]. Study participants presented a mean abdominal circumference of 92.5 cm [SD: 12.6].

#### 3.1.2. Questionnaires

Table 2 provides an overview of the fish consumption habits; on average, less than 10 g of fish were obtained per day from each of the four sources examined (Figure 2). Mean EPA and DHA intake from lean fish was 25 mg/d (8.2 g fish × 3 mg EPA plus DHA/g fish) and from fatty fish was 183 mg/d (12.2 g × 15 mg/g), resulting in an overall intake of 208 mg/d (Table 2).

Thirty-four participants (6.8%) stated that they had taken omega-3 fatty acid supplements regularly within the previous month. The proportion of people taking supplements was the same in men and women (6.8% each). Intake ranged from one capsule per week to two capsules per day. Twenty-five participants took supplements on their own initiative, while 16 indicated general nutritional recommendations, other reasons, and medical recommendations. Questionnaire data indicated approximate daily EPA and DHA intake through supplements to average half a capsule or 0.5 g oil containing 210 mg EPA plus DHA [35]. Adding this additional omega-3 fatty acid intake fraction to the intake of the total baseline population increased the average EPA plus DHA intake at baseline by 15 mg to 221 mg/d (Table 2), which is considerably below the prespecified value of 0.5 g/d (*p* < 0.001).

Thirty-two participants (6.4%) usually followed a vegetarian diet, the proportion being higher in women (9.0%; *n* = 12) than in men (5.5%; *n* = 20). Of those usually following a vegetarian diet, only five indicated not consuming any fish or seafood. No participant followed a vegan diet. At baseline, 20.0% indicated taking advantage of the employee’s canteen service Monday to Friday and about the same number at least several times per week. About one fourth (23.4%) used the canteen once per week to once per month and 33.6% never. 

#### 3.1.3. Blood Parameters

Participants arrived at the health center without the requirement of fasting. Nevertheless, 27.4% indicated having not consumed energy containing food or beverages for at least 5 h before examination and were considered as “fasted” in this study. Median systolic blood pressure (mmHg) was 130 [IQR: 120–142] and diastolic blood pressure (mmHg) was 80 (IQR: 78–90). 

Baseline median plasma lipids (in mg/dL) were: triglycerides 120 (IQR: 82–171), HDL-cholesterol 53 (IQR: 44–63) and LDL-cholesterol 122.5 (IQR: 103–144). The median Lp(a) value at baseline was (in mg/dL) 9.7 (IQR: <LOD-23.6) and slightly lower in men 9.3 (IQR: <LOD-22, 5)) than in women 10.2 (IQR: 6.1–26.8). Median HbA1c (in %) was 5.2 (IQR: 5.1–5.4). Other clinical and biochemical diagnostic parameters were in their normal ranges and are shown in Table A1.

#### 3.1.4. Omega-3 Index

The mean and median O3I at baseline (*n* = 500) were 4.1% (SD: 1.1%) and 3.9% (IQR: 3.3%–4.7%), respectively. Both values thus considerably fell below 8% (*p* < 0.001 for sign test). The median O3I in men was lower than in women (Figure 3, Table 3).

Participants over 55 years of age presented the highest median O3I (median 4.2%, Table 3). This age group also had the lowest share of participants with an O3I below 4%. The proportion of women with an O3I ≥ 4% was higher than that of men (61.0% vs. 41.2%). The median O3I was higher for managerial staff (4.3%) than for skilled/supervisory workers and manual workers (3.8% respectively). The O3I did not differ significantly between employees with rotating shift (3.7%) and employees with day working hours (4.0% each) even after multiple adjustment for all other variables shown in Table 3. In the regression model, the average O3I of shift workers was 0.08 percentage points higher compared to day workers (95%-CI: −0.21; 0.37; *p* = 0.596). With respect to lifestyle-related characteristics, the highest median O3I was found in normal-weight (4.1%) and non-smoking employees (4.0%). The small group of participants indicating vegetarian lifestyle and not to consume fish contributed the lowest O3I in this study (median O3I; 2.7%, Table 3).

The average O3I of participants who consumed any amount of fish in the four weeks prior to taking part in the study was significantly higher than that of participants who did not (Table 4). Looking at categories, most participants who had consumed fatty fish had an O3I above 4%, whereas only 16.7% of those indicating not having consumed fish.

As expected, participants using dietary omega-3 fatty acid supplements had a higher median O3I (4.5%) compared to participants not supplementing omega-3 fatty acids (3.9%). Regular use of the company canteen was another predictor for higher O3I mean and median values, as was the choice of fatty fish on offer in the canteen (Table 4). 

### 3.2. Follow-Up

#### 3.2.1. General Characteristics of Participants in the Follow-Up

Participants willing to follow-up (*n* = 295) differed according to gender, occupational status, working time system, BMI, smoking status, O3I, and visits to the company restaurants. Accordingly, women, employees with occupational status skilled/supervisory workers and managerial staff, day workers (vs. shift workers), normal-weight and non-smoking participants, or participants with an O3I ≥ 4% and participants with regular visits to the company canteen were more willing or able to participate in the follow-up. 

#### 3.2.2. Changes in Food Consumption between Baseline and Follow-Up

In those participating in both phases of the study (*n* = 295), the consumption of lean fish between baseline and follow-up was increased in 106 (36%), reduced in 65 (22%), and left unchanged in 115 (39%) participants, plus 9 missing values. Corresponding values for the consumption of fatty fish were similar. Although the overall median intake of fish did not change regardless of type, the overall mean intake increased. Predictions from mixed modelling showed that consumption of lean fish and fatty fish increased by 1.3 g/d and 3.9 g/d, respectively, resulting in an increase in EPA and DHA intake of 63 mg/d to a total of 266 mg/d (Table 2).

Out of 23 participants with self-indicated omega-3 fatty acid supplementation at baseline, 4 had stopped and 42 had started supplementation during the study. Thus, 61 participants or 21% indicated omega-3 fatty acid supplementation at follow-up. Questionnaire data indicated that the amount of EPA and DHA intake through dietary supplements at follow-up was unchanged compared to baseline with about 210 mg EPA and DHA [35]. This increased the total intake at follow-up through supplements by 30 mg/d to a total EPA and DHA intake of 314 mg or 94 mg/d more than at baseline (Table 2). 

Four of the 22 participants who stated that they were eating mostly vegetarian food at baseline no longer indicated this at follow-up. Conversely, at follow-up, seven participants indicated that they had started to consume a predominantly vegetarian diet. At follow-up, participants reported taking advantage of the increased offer of fatty fish in the canteen. Of those participating at baseline and follow-up, 82, 27, and 181 participants stated having increased, decreased or maintained their consumption of fatty fish in company canteens, respectively (5 missing values excluded).

#### 3.2.3. Changes in O3I between Baseline and Follow-Up

The median O3I was 3.9 at baseline and increased to 4.5% at follow-up (Figure 4).

A total of 204 (69.2%) out of 295 participants for whom baseline and follow-up data were available increased their O3I, 90 people reduced their O3I (30.5%), and one person had the same value at both times (Figure 5).

According to the results of the linear mixed models, average O3I increased significantly by 0.55 (Table 5, values in percentage points) between baseline and follow-up, with highest absolute changes in managerial staff (+0.82), smokers (+1.00), respondents over 45 years, and those visiting the company restaurants several times per week (+0.60) or daily (+0.68). 

Comparing O3I between baseline and follow-up, observable differences in the change were significantly higher in managerial staff compared to manual workers (+0.59) and lower in former smokers compared to smokers (−0.70, Table 5). The respective changes by age, gender, working time system, BMI, vegetarian diet, and eating in company restaurants did not differ. 

## 4. Discussion

This study of employees of a large German company explored the omega-3 fatty acid intake in the form of EPA- and DHA-containing food and dietary supplements, the employees’ omega-3 fatty acid status by O3I, and changes in the O3I upon recommendation of higher intake to those with low O3I status. At baseline, the O3I correlated with health and demographic parameters, but not with work shift rotations. The recommendation to increase omega-3 fatty acid intake through the diet and an improved offering of fatty seafood by the company canteen increased median intake and O3I in study participants modestly. 

### 4.1. Low Omega-3 Fatty Acid Intake in the Southern German Working Population

Study participants consumed on average 21.6 g/d of EPA- and DHA-rich foods. Highest contributors were fatty and lean fish (mean 20.4 g/d, Table 2), whereas omega-3 fatty acid-enriched eggs or milk (mean 1.2 g/d) contributed only modest amounts. Intakes in our study were somewhat lower than intakes estimated for the average German adult population with a total intake of fish and fish products of 23 g/d for women and 29 g/d for men, respectively [36]. Fish intake for a representative sample of the European populations was recorded between 1990 and 1999 in the EPIC cohort. German daily fish consumption averaged 19 g for men and 15 g for women [33]. Total fish intake for the adult US population was estimated to be 25 g/d [37]; however, the proportion of fatty fish was lower than in our German working population. Intake of fish and fish products including marginal intake from omega-3 fatty acid-enriched foods recorded in this study contributed dietary EPA and DHA of 208 mg/d at baseline. Fatty fish was not only in this study the primary contributor of EPA and DHA, but also for the US population [37]. 

Estimates of the number of adult individuals consuming dietary omega-3 fatty acid supplements vary greatly. Data from the US National Health and Nutrition Examination Survey (NHANES) showed that in the time period between 1999 and 2012, dietary fish oil supplement intake increased from 1.3% to 12% [38]. Recent results from the NHANES survey indicated that EPA and DHA dietary supplement intake within the previous 30 days was only about 1% [37]. Moreover, Jackson et al. [39] reported that about half of subjects who self-tested their O3I took supplements. In our study, 6.8% of participants indicated omega-3 fatty acid supplement use during the 28 days prior to baseline.

### 4.2. Low Baseline O3I Reveals Typical Characteristics

The mean O3I at baseline of our participants was surprisingly low (4.1%, Table 3) when comparing studies investigating other cohorts in Germany. In the Ludwigshafen Risk in Cardiovascular Health (LURIC) study investigators found higher average O3I values and O3I inversely correlating with mortality independent of other risk factors [28] The non-coronary artery disease reference cohort of LURIC included 721 male and female subjects, was older than participants in our study (age 58.6 ± 12.1 years vs. 44.8 ± 10.7 years, mean ± SD) and showed a higher O3I (5.7 ± 1.2% vs. 4.1 ± 1.1%, mean ± SD). A higher O3I (5.5%) was also measured in 100 healthy German male and female adults (age 40 ± 14 years, mean ± SD) [40]. In the German VitaMinFemin study, the O3I was assessed for a subset of the middle-aged women (*n* = 446). The average O3I was 5.5% and almost 88% of the women were in the 4–8% O3I range [41], whereas only 61% of women in the current study had an O3I ≥4%. These three studies and our study used the same analytical method for O3I assessment [13].

The increase in O3I values with age observed in this study (medians: 3.8% for <35 y to 4.2% for ≥55 y of age) is consistent with previous observations [15,41,42,43]. The LURIC investigators observed a significant increase in age along tertiles of DHA, although there was no relationship between EPA and age in this predominantly (70%) male cohort [42]. Positive relationships between EPA, DHA and O3I values with age were also observed in a large de-identified and aggregated dataset from a US laboratory [15]. The study showed that differences in O3I may be attributed to age with EPA and DHA erythrocyte concentrations increasing from the 1st until the 7th decade of life. In the subset of VitaMinFemin, for which O3I was determined, the older half of the women presented higher EPA and O3I concentrations, whereas the effects for DHA were not significant [41]. 

Gender differences were observed in our study and previously. Female employees in our study had slightly higher mean O3I (4.4%) than their male colleagues (4.0%), but the mean was still more than one percentage point lower than in the female VitaMinFemin participants [41]. A gender difference was also observed in the cross-sectional study of U.S. and German adults with an at least one percentage point higher O3I in women than in men [40]. In LURIC, a gender difference could not be detected (all participants) [42]. Data from the large de-identified US study found that mean EPA levels were reduced in women compared to men through their 40s, whereas DHA was higher in women in their teens and 20s [15]. This could indicate an important role of DHA during women’s fertile years [2]. Thus, the gender and age differences in O3I appear to be consistent. They are reflected in our employee cohort even though absolute O3I tended to be lower than reported previously.

Occupational status was a factor associating with O3I. Manual and skilled/supervisory workers had O3I values significantly lower than managerial staff suggesting a social gradient. Background knowledge on health and socioeconomic status in study participants may have affected the difference. Managerial staff with interest in the study may have had more knowledge about and more interest in omega-3 fatty acid-rich diets than their colleagues.

Work in rotating shifts apparently did not affect O3I values. We consider this as a positive outcome of the study. Previous work found a marginal decline in systolic blood pressure and an elevation of triglycerides related to shift work, but no difference with respect to other cardiovascular risk factors [44]. The current study did not show differences between employees on rotating vs. day shifts in O3I, cholesterol or triglyceride concentrations associated shift work schedules.

Lifestyle appeared to affect O3I in our study. Higher O3I was associated with normal rather than obese BMI (median 4.1% vs. 3.8%), and with non-smoking rather than with current smoking status (median 4.5% vs. 3.9%). These among several additional modifiable health associated markers [45] may reflect the level of health consciousness and lifestyle of the employees. However, in our employees, O3I was not associated with the metabolic health markers of central obesity, hypertension, or HbA_1C_ (data not shown). Another lifestyle factor appears to be the attendance of the company canteen. Regular guests had half a percentage point higher O3I than those indicating never having attended. 

Vegetarian lifestyle, defined as not consuming meat or fish products, usually contain insignificant dietary sources of EPA and DHA. In our study, however, 84% of participants indicating to follow a predominantly vegetarian lifestyle, indicated intake of some fish. Therefore, it is not surprising that the O3I of vegetarians consuming fish did not differ from the overall population. In our study, vegetarians not consuming fish had a very low median O3I of 2.7%. This seemed unexpectedly low, because vegans do not necessarily have lower O3I than omnivorous subjects, both consuming insignificant amounts of omega-3 fatty acids [43]. 

Intake of dietary omega-3 fatty acid supplements in our study was estimated to contribute 210 mg/d of EPA and DHA. The median O3I of participants taking omega-3 fatty acid supplements (4.5%) was 0.6 percentage points higher than those not supplementing (3.9%). Efficacy of dietary EPA and DHA supplementation on O3I can vary widely depending on dose, source, chemical form of the fatty acids, formulation, frequency of intake, duration of intervention, competing fatty acids, gender, BMI, genetics of the individual and other factors [17,18]. Walker et al. [18] recently reviewed studies supplementing EPA and DHA in order to predict effects. Assuming a mean baseline O3I of 4%, they modeled that an increase of 0.6 percentage points would be achieved on average by adding 225 mg EPA and DHA daily. Similarly, McDonnell et al. [46] estimated that in non-fish eaters, a 0.6 percentage point increase in O3I would require 230 mg daily. This corresponds well with the difference between supplementing and non-supplementing study participants in the present study.

### 4.3. Moderate Effects of Dietary Recommendations

Occupational health nutrition studies come with constraints including that control groups may not be feasible, the limited possibility to incentivize full study participation, and restrictions in performing interventions associated with commercial products. Therefore, this study was designed as an open employee survey to generate baseline data on the omega-3 fatty acid status and to explore the possibility of affecting the O3I by a recommendation to increase dietary intake of omega-3 fatty acids for health. 

At baseline, study participants were informed about the potential cardiovascular health benefits of increasing dietary intake of omega-3 fatty acids by their physician and using an information graphic. The company canteen services increased their offer of fatty fish to two per week. The awareness of the importance of omega-3 fatty acids motivated participants to increase their calculated daily intake of EPA and DHA by about 100 mg from fish and omega-3 fatty acid supplements. This additional EPA and DHA intake adds up to 700 mg per week or one serving of 4 oz (113 g) mixed fish with more lean than fatty fish [47]. The successful increase in fish intake was also seen in 26% of participants moving up one O3I category (from <4% to 4–8% or from 4–8% to >8%) vs. 7% reducing one category (Figure 5). This may in part be attributed to the more frequent visit of the company canteen and its more generous offer of fatty fish. Awareness of the role of EPA and DHA contributed to some participants starting supplementation of omega-3 fatty acids.

The increased consumption of EPA and DHA in our study resulted in a median and mean O3I increase of 0.5 percentage points in those completing the four-months study period and related to 100 mg/d additional EPA and DHA intake. In a cross-sectional study with predominantly US and Canadian participants (*n* = 985), a median increase of 0.5% in O3I corresponded to one additional serving of fish per week [46]. Several studies assessed the efficacy of EPA and DHA intake from food or supplements starting O3I in a similarly low range as in our study. The equivalent to 100 mg/d additional EPA and DHA intake in an eight-week study with 14 participants starting with 4.7% O3I achieved a 0.2 percentage point increase in O3I [48]. A higher efficacy of a 0.6 percentage point O3I increase per 100 mg/d over eight weeks was achieved in 22 subjects, starting at 4.2%. Starting from an even lower O3I at baseline of 3.1% in *n* = 46 participants, an 0.7 percentage point increase was achieved per 100 mg/d over four months [43], whereas in a study of 24 subjects starting from a O3I of 4.2%, a 0.6 percentage point increase was achieved per 100 mg/d after 5 months [49]. Fish and fish oil supplements were tested as a dietary source of EPA and DHA in a small study (*n* = 23), starting with an O3I of 4.0–4.3%. Both dietary sources achieved an equal increase of 0.4 percentage points per 100 mg/d [50]. Thus, efficacies of low dose dietary EPA and DHA supplements affecting O3I varies [48]. 

Comparing baseline and follow-up in the present study, the age-associated O3I increase and a higher O3I for females were retained, but the delta remained unaffected. Moreover, effects on the O3I delta between timepoints were absent for work system, BMI, smoking status, exercise level and the parameters recorded to assess metabolic syndrome. Effects on metabolic health were not expected due to the only small increase achieved through dietary recommendation and the large individual variability in response to increased EPA and DHA intake typically observed [48]. 

### 4.4. Omega-3 Fatty Acid Status for Employee Health Campaigns

The assessment of the omega-3 status in an occupational health environment is difficult to establish due to ethical and data protection regulations, but could add a useful parameter related to cardiovascular and metabolic health. Erythrocyte long-chain omega-3 fatty acid concentrations are inversely associated with mortality and with incident cardiovascular disease [28,51] and a low O3I is associated with early-onset coronary atherosclerosis [52,53]. Moreover, the evidence that higher circulating EPA and DHA concentrations are associated with a lower metabolic syndrome risk [54] suggests that a higher O3I might have also benefits in the work environment. 

Using the O3I as parameter in occupational health assessments is attractive because it can be modified by increasing EPA and DHA intake through foods or dietary supplements and with lifestyle. A risk reduction was observed for habitual users of fish oil supplements versus non-users with estimated risk reductions for all-cause mortality (13%), CVD mortality (16%) and CVD events (7%) [25]. However, non-modifiable factors such as lifestyle, gender, age and genetics also affect the O3I [10,15].

The O3I has been shown to be low in our and other studies in southern Germany [42,55]. This should motivate the consideration of O3I as a biomarker indicating cardiovascular or metabolic risk and considering whether achieving higher O3I may be a reasonable target. In the trusted relationship between the occupational health service provider and employee, small effects were achieved through information about the benefits of higher intake of omega-3 fatty acids in this study. This supports the concept of recommendations for higher intake of omega-3 fatty acids and the O3I as biomarker in an occupational health setting. We suggest that greater increases in the O3I might be achievable by extending the observation period and by intensifying dietary counselling. Another open question is whether, in the background of the general Western lifestyle, supplementation should receive stronger emphasis.

### 4.5. Limitations

The subgroup of employees registering for our study reflected the overall company demographics, almost 33,000 employees at the Ludwigshafen site, with respect to age, gender, occupational status and work shift distribution (Table 1). There was, however, a slightly lower relative study participation of men, manual workers, and rotating shift employees (Table 1). This might indicate a lower health interest in these subgroups and could have affected study results in favor of an over-estimation of the effect of our intervention,

Moreover, results are limited by the pre-post pilot study design. This does not allow conclusions regarding potential causal relationships between intake of omega-3 fatty acids and independent variables. Furthermore, the possibility of selection bias must be recognized. During the study recruitment of about two months, 1162 employees were invited to participate in the study. Of those, 509 (44%) accepted for the baseline survey, of which 500 fulfilled the inclusion criteria and 295 successfully completed the follow-up. Thus, 59% of participants included in the study completed the follow-up. Motivation to participate in the study and specifically the follow-up may have been biased to those with somewhat higher interest in a healthy diet and lifestyle, whereas employees with worse perceived health or those who felt inconvenienced by the second sampling could not be encouraged to follow-up. However, considering baseline participation, the distribution of study participants compared reasonably well with the employee demographics at the company’s Ludwigshafen site data from 2018 with respect to age, gender, occupational status and work shift system. In addition, baseline intake of EPA and DHA from foods and supplements did not differ between all included participants (221 mg/d) and those that chose to participate in both study parts (215 mg/d).

Depending on whether an imbalance was introduced by self-selection of participants, the level of the O3I could be under- or overestimated in our sample. However, considering that a bias would be expected towards the more health-conscious participants with higher omega-3 fatty acid status, baseline O3I was rather low, even compared to other studies in southern Germany [42,55]. Furthermore, information on the intake of omega-3 fatty acid-rich food and dietary supplements relies entirely on questionnaires filled out by the employees. Questionnaires can be subjective, but the one used was part of a validated food intake questionnaire and belongs to the best tools currently available to study food intake in Germany [36]. 

Employees on day shifts received an improved offer of fatty fish through the company-wide canteen system. This may have introduced bias against rotating shift workers who had less frequent access to the canteens. Moreover, a control group could not be included in the study design for practical and ethical reasons. Restricting the improved canteen offers to only one study group would have introduced significant changes to their normal work environment and thus additional confounders. From an ethical perspective, it was not reasonable to include a control group. Control participants would have been asked for an additional blood draw (follow-up) without apparent study benefits. A further limitation which should be acknowledged is the possibility for regression towards the mean, meaning that outlier observations at baseline tend to be followed by values that are closer to the average values at follow-up. Finally, it should be noted that a variety of statistical tests were carried out, potentially increasing the possibility for a type I error (rejection of a true null hypothesis). Due to the predominantly exploratory character of our analyses, we did, however, not formally adjust for multiple testing.

## 5. Conclusions

In this occupational health study of a large German company, we found that employees presented a median O3I of 3.9%, which was associated with an estimated intake of 221 mg/d EPA and DHA predominantly from fatty fish. Three fifths of the cohort also participated in the follow-up sampling. Recommendation to increase intake of omega-3 fatty acids for health reasons resulted in an increased intake of about 100 mg/d EPA and DHA and a 0.5% elevation of the O3I after 4 months. This study presents essential estimates of blood concentrations of omega-3 fatty acids in a southern German working population and the challenges of increasing the O3I even in study participants motivated to a follow-up on their omega-3 fatty acid status. The usefulness of the O3I in employee health assessment needs further consideration.

## Figures and Tables

**Figure 1 nutrients-12-03267-f001:**
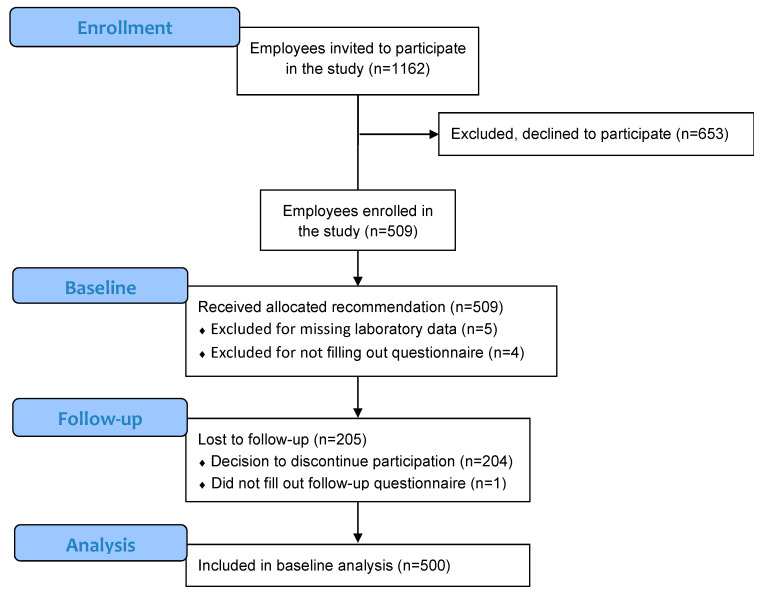
Flow diagram for study design and flow chart of recruitment, baseline (October to December 2017) and follow-up (March to April 2018) analysis.

**Figure 2 nutrients-12-03267-f002:**
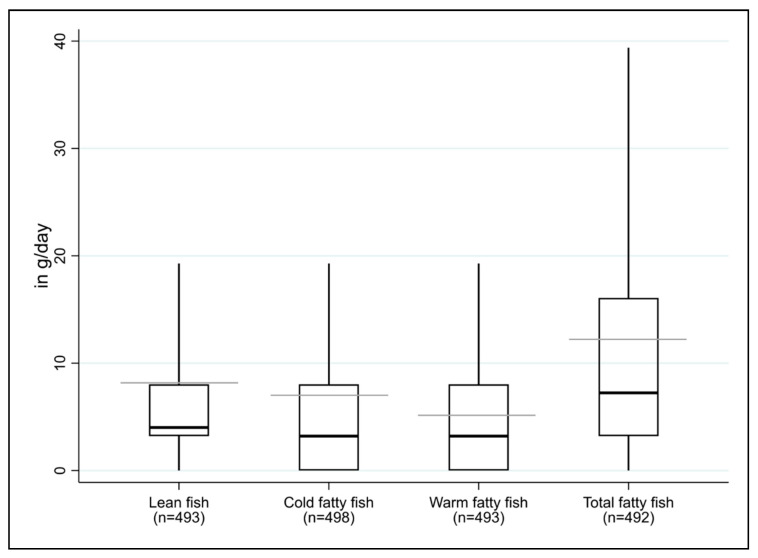
Distribution of baseline consumption of total lean and fatty fish (cold/warm; in g/day) by study participants. Box plot of the distribution of the consumption of lean and warm/cold fatty fish (in g/day) for all participants at baseline. The gray horizontal line shows the arithmetic mean, the bold, black horizontal line inside the box shows the median. The bottom and top of the box are the bottom and top quartiles of the distribution. The length of the black, vertical “whiskers” corresponds to the 5th and 95th percentile of the variable.

**Figure 3 nutrients-12-03267-f003:**
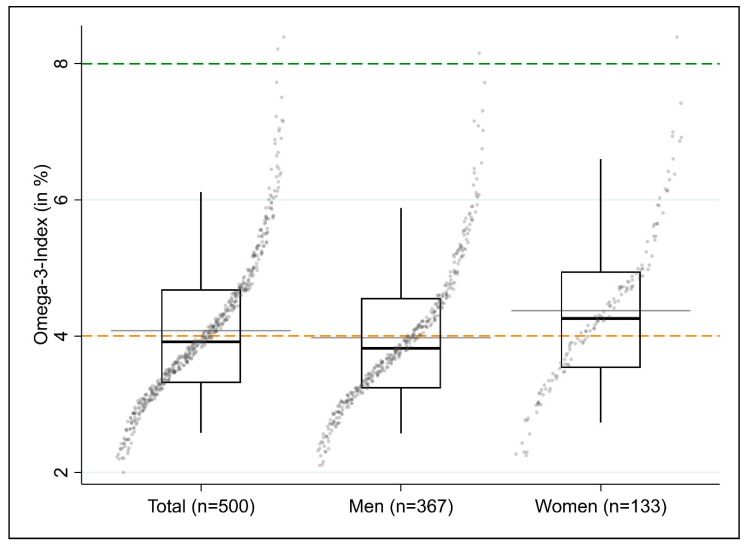
Box quantile plot of the distribution of the omega-3 index (in %) for all participants at baseline (*n* = 500) and stratified by gender. The gray horizontal line shows the arithmetic mean, and the bold, black horizontal line inside the box shows the median. The bottom and top of the box are the bottom and top quartiles of the distribution. The length of the black, vertical “whiskers” corresponds to the 5th and 95th percentile of the variable. The quantile plot plots the ordered values of the respective variables (ordinate) against the cumulative probability (abscissa). The dashed horizontal lines each show the borders between the levels of cardioprotection, largest to average (green, 8%) and average to least (orange, 4%). For illustrative purposes, horizonal dispersion (jitter) was added to the individual datapoints.

**Figure 4 nutrients-12-03267-f004:**
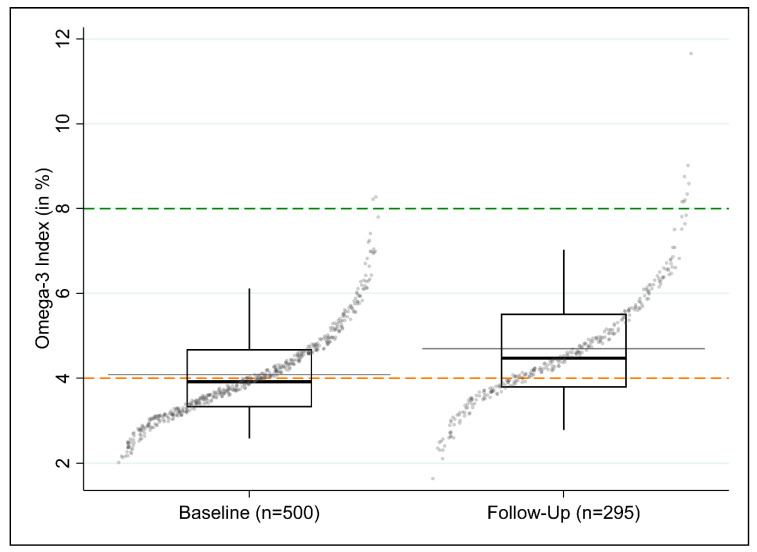
Box quantile plot for the distribution of the O3I at baseline (*n* = 500) and follow-up (*n* = 295) of the study. The gray horizontal line shows the arithmetic mean, the bold, black horizontal line inside the box shows the median. The bottom and top of the box are the bottom and top quartiles of the distribution. The length of the black, vertical “whiskers” corresponds to the 5th and 95th percentile of the variable. The quantile plots the ordered values of the respective variables (ordinate) against the cumulative probability (abscissa). For illustrative purposes, horizonal dispersion (“jitter”) was added to the individual datapoints.

**Figure 5 nutrients-12-03267-f005:**
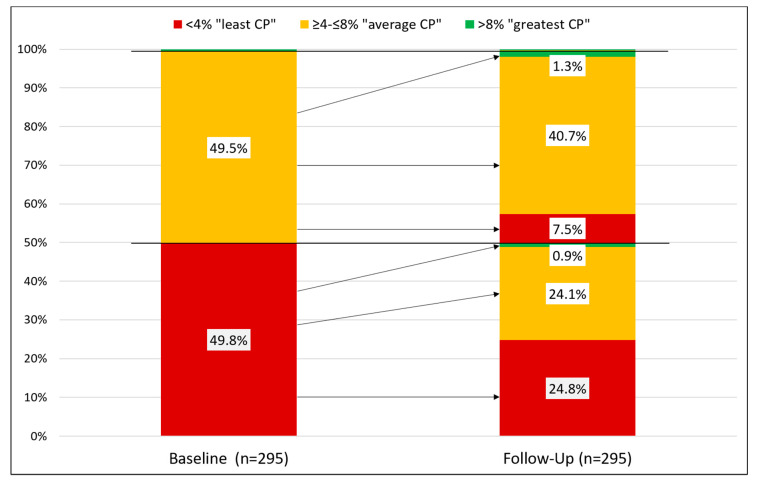
Bar chart to show the change in the categorical omega-3 index between baseline and follow-up (*n* = 295). Relative frequencies <5% are not labeled in the illustration for reasons of legibility; due to rounding, the sum of the percentages may deviate from 100%. Arrows indicate the group change from baseline to follow-up.

**Table 1 nutrients-12-03267-t001:** Sociodemographic, occupational and lifestyle-related factors of participants in the Omega-3 Employee Study at baseline compared to the overall company staff at the BASF Ludwigshafen site on 1st January 2019.

Omega-3 Employee Study (Baseline)							Company Population on 1st January 2019					
	**Total**		**Male**		**Female**		**Total**		**Male**		**Female**	
	*n*	% ^a^	*n*	% ^a^	*n*	% ^a^	*N*	% ^a^	*n*	% ^a^	*n*	% ^a^
**Total**	500	100	367	100	133	100	32,921	100	25,909	100	7012	100
**Age (in years)**												
<35	100	20.0	66	18.0	34	25.6	6561	19.9	4787	18.5	1774	25.4
35–39	52	10.4	33	9.0	19	14.3	3238	9.8	2348	9.1	890	12.7
40–44	69	13.8	52	14.2	17	12.8	3542	10.8	2615	10.1	927	13.2
45–49	76	15.2	55	15.0	21	15.8	4905	14.9	3695	14.3	1210	17.3
50–54	101	20.2	79	21.5	22	16.5	6299	19.1	5125	19.8	1174	16.7
≥55	102	20.4	82	22.3	20	15.0	8376	25.4	7339	28.3	1037	14.8
**Gender ****												
Male	367	73.4	-	-	-	-	25,909	78.7	-	-	-	-
Female	133	26.6	-	-	-	-	7012	21.3	-	-	-	-
**Occupational status ***^,b^**												
Manual worker	109	21.8	106	28.9	3	2.3	10,337	31.4	9762	37.7	575	8.2
Skilled/supervisory worker	250	50.0	158	43.1	92	69.2	13,917	42.3	9653	37.3	4264	60.8
Managerial staff	140	28.0	103	28.1	37	27.8	8667	26.4	6494	25.1	2173	31.0
**Working time system ***^,c^**												
Day work	415	83.0	285	77.7	130	97.7	24,997	75.9	18,195	70.2	6802	97.0
Shift work	85	17.0	82	22.3	3	2.3	7915	24.0	7710	29.8	205	2.9
**Body mass index (in kg/m^2^)**												
Normal weight/underweight (<25)	203	40.6	115	31.3	88	66.2						
Overweight (25–30)	220	44.0	195	53.1	25	18.8						
Obesity (≥30)	74	14.8	57	15.5	17	12.8						
**Smoking status**												
Smoker (cigarette/e-cigarette)	60	12.0	49	13.3	11	8.3						
Former smoker	117	23.4	94	25.6	23	17.3						
Nonsmoker	320	64.0	224	61.0	96	72.2						

^a^ Presented percentage values are column percentages; ^b^ A trainee (Omega-3 study) was not taken into account; ^c^ Respectively, 2 persons (Omega-3 study) and 67 persons (company population) with “telework” were counted as “day work”; people with missing values for BMI (*n* = 3) and smoking status (*n* = 3) are not listed in the table; *** *p* < 0.001, ** *p* < 0.01 for Pearson’s chi-squared test.

**Table 2 nutrients-12-03267-t002:** Intake of fish and dietary supplements from dietary questionnaire and intake of eicosapentaenoic acid (EPA) + docosahexaenoic acid (DHA) derived thereof.

	Baseline (*n* = 500)			Follow-Up (*n* = 295)			Baseline (*n* = 295)			Predicted within Difference between Baseline and Follow-up and 95%-CI ^†^	
	*n*	Fish	EPA + DHA	*n*	Fish	EPA + DHA	N	Fish	EPA + DHA	Fish	EPA + DHA
	(%) ^a^	(g/d) ^b^	(mg/d) ^c^	(%) ^a^	(g/d) ^b^	(mg/d) ^c^	(%) ^a^	(g/d) ^b^	(mg/d) ^c^	(g/d)	(mg/d)
Lean fish	493 (82.2)	8.2 (19.7)	24.5 (59.2)	290 (85.1)	8.7 (9.5)	26.1 (28.4)	291 (82.4)	6.8 (6.6)	20.5 (20.0)	1.3 (0.1–2.5)	4.0 (0.4–7.6)
Fatty fish	492 (79.2)	12.2 (15.6)	183.4 (233.7)	291 (84.8)	16.0 (19.9)	240.2 (298.2)	291 (79.0)	11.9 (15.2)	178.7 (SD: 227.3)	3.9 (1.6–6.2)	58.7 (24.3–93.1)
Omega-3 fatty acid supplements	485 (6.8)	-	14.7 (53.7)	291 (22.0)	-	46.9 (87.6)	288 (7.8)	-	16.8 (57.0)	-	30.3 (21.1–39.4)
Total		20.4 (28.6)	220.7 (269.3)		24.8 (24.5)	314.3 (317.8)		18.8 (19.2)	215.3 (249.8)	5.3 (2.5–8.1)	94.1 (56.5–131.7)

^a^ Participants responded with positive intake to the question over total respondents; ^b^ mean (standard deviation) are presented (except where indicated) for comparability with the scientific literature; ^c^ EPA + DHA content for low and high fat fish species was estimated from the US Nutrient Content database [34], assuming that 1 g of lean fish contains 3 mg, and 1 g of fatty fish contains 15 mg of EPA and DHA; EPA and DHA intake through supplements averaged half a capsule or 0.5 g oil containing 210 mg EPA and DHA. ^†^ Predictions based on linear mixed models.

**Table 3 nutrients-12-03267-t003:** Omega-3 index (categorical and continuous) stratified according to selected sociodemographic, occupational and lifestyle-related factors (*n* = 500).

		Omega-3 Index					
		Continuous		Categorical			Multivariable Linear Regression Analysis
	*n*	Mean (SD)	Median (IQR)	<4% (Least CP) % ^a^	4%–8% (Average CP) % ^a^	>8% (Largest CP) % ^a^	Coeff. (95% CI)
**Total**	500	4.1 (1.1)	3.9 (3.3–4.7)	53.6	46.0	0.4	*n* = 484
**Age (in years)**							
<35	100	4.0 (1.0)	3.8 (3.2–4.5)	60.0	40.0	0.0	−0.01 (−0.33; 0.32)
35–39	52	4.1 (0.9)	3.8 (3.3–4.6)	57.7	42.3	0.0	0.04 (−0.33; 0.42)
40–44	69	4.0 (1.0)	3.8 (3.2–4.6)	58.0	42.0	0.0	Reference
45–49	76	4.0 (1.0)	4.0 (3.3–4.7)	54.0	46.1	0.0	0.09 (−0.25; 0.43)
50–54	101	4.2 (1.3)	3.9 (3.4–4.8)	52.5	45.5	2.0	0.42 (0.09; 0.75)
≥55	102	4.2 (1.1)	4.2 (3.3–4.8)	43.1	56.9	0.0	0.35 (0.03; 0.68)
**Gender *****							
Male	367	4.0 (1.0)	3.8 (3.2–4.6)	58.9	40.9	0.3	Reference
Female	133	4.4 (1.2)	4.3 (3.5–5.0)	39.1	60.2	0.8	0.51 (0.27; 0.74)
**Occupational status ***^,a^**							
Manual worker	109	3.9 (1.0)	3.8 (3.2–4.3)	57.8	42.2	0.0	Reference
Skilled/supervisory worker	250	3.9 (1.0)	3.8 (3.2–4.5)	59.2	40.8	0.0	−0.23 (−0.50; 0.04)
Managerial staff	140	4.5 (1.2)	4.3 (3.7–5.2)	40.0	58.6	1.4	0.33 (−0.01; 0.66)
**Working time system ^b^**							
Day work	415	4.1 (1.1)	4.0 (3.3–4.7)	51.6	48.0	0.5	Reference
Shift work	85	3.9 (1.0)	3.7 (3.2–4.3)	63.5	36.5	0.0	0.08 (−0.21; 0.37)
**Body mass index (in kg/m^2^) ***							
Normal weight/underweight (<25)	203	4.2 (1.2)	4.1 (3.4–4.9)	48.3	50.7	1.0	Reference
Overweight (25–30)	220	4.0 (1.0)	3.9 (3.3–4.5)	56.8	43.2	0.0	0.04 (−0.18; 0.27)
Obesity (≥30)	74	3.9 (1.1)	3.8 (3.1–4.6)	59.5	40.5	0.0	−0.16 (−0.46; 0.14)
**Smoking status *****							
Nonsmoker	320	4.2 (1.1)	4.0 (3.4–4.8)	50.6	48.8	0.6	Reference
Former smoker	117	4.1 (1.1)	3.9 (3.4–4.6)	52.1	47.9	0.0	0.01 (−0.22; 0.25)
Smoker (cigarette/e-cigarette)	60	3.6 (0.8)	3.5 (3.0–4.1)	73.3	26.7	0.0	−0.32 (−0.63; −0.01)
**Vegetarian diet ****							
Strict (no fish)	5	2.6 (0.3)	2.7 (2.3–2.9)	100.0	0.0	0.0	−1.62 (−2.54; 0.70)
Predominantly	27	4.2 (0.9)	4.3 (3.4–4.7)	37.0	63.0	0.0	−0.05 (−0.45; 0.36)
No	466	4.1 (1.1)	3.9 (3.3–4.7)	53.9	45.7	0.4	Reference
**Eating in company restaurants ****							
Never	168	3.8 (1.0)	3.7 (3.1–4.4)	62.5	37.5	0.0	Reference
One time per month	43	4.0 (1.2)	3.7 (3.1–4.7)	67.4	32.6	0.0	0.16 (−0.20; 0.51)
2–3 times per month	35	4.1 (1.1)	4.0 (3.2–4.6)	54.3	45.7	0.0	0.24 (−0.15; 0.63)
1 time per week	41	4.2 (0.9)	4.1 (3.6–4.7)	46.3	53.7	0.0	0.28 (−0.09; 0.65)
Several times per week	103	4.3 (1.1)	4.1 (3.5–4.9)	45.6	53.4	1.0	0.27 (−0.02; 0.56)
Daily (Monday-Friday)	100	4.3 (1.1)	4.1 (3.5–4.8)	44.0	55.0	1.0	0.25 (−0.05; 0.55)

SD: standard deviation; IQR: interquartile range; CP: cardioprotection; ^a^ presented percentage values are row percentages; ^b^ a trainee was not taken into account; (c) 2 persons with “telework” were counted as “day work”; people with missing values for BMI (*n* = 3), smoking status (*n* = 3), vegetarian diet (*n* = 2), and eating in company restaurants (*n* = 10) are not listed in the table and were not taken into account when calculating the statistical tests; *** *p* < 0.001, ** *p* < 0.01, * *p* < 0.05 for Mann–Whitney U test (2 categories) or Kruskal–Wallis test (>2 categories).

**Table 4 nutrients-12-03267-t004:** Omega-3 index (categorical and continuous) stratified according to consumption of lean and fatty fish, diet, supplementation and consumption of fish in the canteens (*n* = 500).

		Omega-3-Index (in %)				
		*Continuous*		*Categorical*		
	*n*	Mean (SD)	Median (IQR)	<4% (least CP) % ^a^	4%–8% (average CP) % ^a^	>8% (largest CP) % ^a^
**Consumption of lean fish (g/d) *****						
Yes	411	4.2 (1.0)	4.0 (3.5–4.7)	49.2	50.4	0.5
No (=0 g/day)	82	3.5 (1.0)	3.3 (2.7–4.0)	78.1	22.0	0.0
**Consumption of fatty fish (g/d) *****						
Yes	396	4.3 (1.0)	4.1 (3.5–4.8)	46.7	52.8	0.5
No (=0 g/d)	96	3.3 (0.9)	3.2 (2.7–3.8)	83.3	16.7	0.0
**Omega-3 supplementation *****						
Yes	34	4.9 (1.4)	4.5 (3.9–5.8)	35.3	61.8	2.9
No	451	4.0 (1.0)	3.9 (3.3–4.7)	55.4	44.4	0.2
**Consumption of fatty fish in canteens of the company ***^,a^**						
Never	312	3.9 (1.0)	3.7 (3.2–4.4)	61.5	38.5	0.0
One time per month	118	4.4 (1.0)	4.2 (3.5–5.0)	40.7	59.3	0.0
2–3 times per month	46	4.5 (1.3)	4.3 (3.6–5.0)	39.1	56.5	4.4
1 time per week	19	4.5 (1.0)	4.8 (3.7–5.2)	31.6	68.4	0.0

SD: standard deviation; IQR: interquartile range CP: Cardioprotection; ^a^ presented percentage values are row percentages; Missing information on consumption of lean-fat fish (*n* = 7), consumption of fatty fish (*n* = 8), omega-3 fatty acid supplementation (*n* = 15), eating in company restaurants (*n* = 10), consumption of fatty fish in company restaurants (*n* = 5) are not listed in the table and were not taken into account when calculating the statistical tests; *** *p* < 0.001, for Mann–Whitney U test (2 categories) or Kruskal-Wallis test (>2 categories).

**Table 5 nutrients-12-03267-t005:** Omega-3 index changes between baseline and follow-up within and between selected sociodemographic and occupation-related groups using linear mixed models.

	Estimated within Group Change of O3I between Baseline and Follow-up Coeff. (95%-CI)	Estimated between Group Differences of Changes in O3I between Baseline and Follow-up Coeff. (95%-CI)
**Variables at baseline**		
**Total**	0.55 (0.42; 0.68)	-
**Age (in years)**		
<35	0.38 (0.09; 0.67)	−0.26 (−0.71; 0.20)
35–39	0.38 (−0.08; 0.85)	−0.26 (−0.84; 0.33)
40–44	0.64 (0.29; 0.99)	*Reference*
45–49	0.70 (0.39; 1.01)	0.06 (−0.41; 0.53)
50–54	0.47 (0.19; 0.75)	−0.17 (−0.62; 0.28)
≥55	0.70 (0.39; 1.01)	0.06 (−0.41; 0.53)
**Gender**		
Male	0.55 (0.39; 0.71)	*Reference*
Female	0.54 (0.31; 0.78)	0 (−0.29; 0.28)
**Occupational status ^a^**		
Manual worker	0.23 (−0.11; 0.57)	*Reference*
Skilled/supervisory worker	0.47 (0.30; 0.65)	0.24 (−0.14; 0.62)
Managerial staff	0.82 (0.59; 1.05)	0.59 (0.18; 1.00)
**Working time system ^b^**		
Day work	0.55 (0.41; 0.69)	*Reference*
Shift work	0.52 (0.11; 0.94)	−0.03 (−0.47; 0.41)
**Body mass index (in kg/m^2^)**		
Normal weight/underweight (<25)	0.61 (0.41; 0.80)	*Reference*
Overweight (25–30)	0.49 (0.28; 0.70)	−0.11 (−0.40; 0.17)
Obesity (≥30)	0.51 (0.17; 0.85)	−0.09 (−0.49; 0.30)
**Smoking status**		
Nonsmoker	0.56 (0.40; 0.71)	*Reference*
Former smoker	0.31 (0.02; 0.59)	−0.25 (−0.57; 0.07)
Smoker	1.00 (0.56; 1.44)	0.45 (−0.02; 0.91)
**Vegetarian diet**		
Strict (no fish)	0.27 (−0.85; 1.39)	−0.28 (−1.41; 0.85)
Predominantly	0.36 (−0.16; 0.89)	−0.19 (−0.73; 0.35)
No	0.55 (0.42; 0.68)	*Reference*
**Eating in company restaurants**		
Never	0.55 (0.29; 0.80)	*Reference*
Once per month	0.33 (−0.12; 0.77)	−0.22 (−0.73; 0.29)
2–3 times per month	0.36 (−0.13; 0.86)	−0.18 (−0.74; 0.37)
1 time per week	0.40 (−0.07; 0.86)	−0.15 (−0.68; 0.38)
Several times per week	0.60 (0.33; 0.88)	0.06 (−0.31; 0.43)
Daily (Monday to Friday)	0.68 (0.41; 0.95)	0.13 (−0.24; 0.51)

Values in percentage points, ^a^ one trainee was not considered; ^b^ two participants with “telework” were counted as “day work”.

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
