# Peer review of "Changes in Erythrocyte Omega-3 Fatty Acids in German Employees upon Dietary Advice by Corporate Health"

_nutrients, 2020, doi:10.3390/nu12113267_

Round 1

Reviewer 1 Report

This health study considers the omega-3 index in a large sample of clinically healthy German workers before and after they were given a recommendation to increase their intake of omega-3 fatty acid rich food. The authors reported a modest increase intake of about 100 mg/d EPA and DHA and a 0.5% elevation of the omega-3 index after 4 months.

This is a very good study. It provides important omega-3 blood data for a large sample of individuals. It also demonstrates the difficulty of changing people's eating habits. 

One minor issue. The results of Table 2 are presented in a footnote for Table 1 (page 10). The results of Table 2 should be presented in a distinct, separate paragraph.  

Author Response

Comment 1:

One minor issue. The results of Table 2 are presented in a footnote for Table 1 (page 10). The results of Table 2 should be presented in a distinct, separate paragraph.

Response:

The reviewer contributed an important advice. Unfortunately, parts of Section 3.1.4 were accidentally moved into the footnote of Table 1 during formatting. We accordingly moved the two sentences back to their original place in Section 3.1.4. and shortened them.

Reviewer 2 Report

I have read the manuscript: ‘Changes in Erythrocyte Omega-3 fatty Acids in German Employees upon Dietary Advice by Corporate Health. In this manuscripts the authors describe a study executed at the BASF corporation; participants received an health check including an assessment of their O3I. If their O3I was below the recommended level they were given advice how to increase their O3I.

My major concern with the current study is the fact that it is not clear what the news value of the current study and that is unclear what the ‘intervention’ entailed exactly. There is a multitude of studies investigating the fish consumption and O3I of people from diverse population as you mention in the discussion; what does your study add? This should be specified in the introduction and at the beginning of the discussion. Additionally, you mention in the methods that people receive a consultation about health benefits of higher O3I and about the cut-offs, and that they received a handout. Additionally, you mention somewhere (rather briefly) that the amount of fish served in the canteen was increased. I feel if you want to discuss the follow up and ‘the effect of your intervention’ you should elaborate more on the intervention aspects. Especially as this is one of your hypotheses (hypothesis 3). In general there is so much information in the manuscript, that a specific focus is lacking.

Comments

Line 81-84: It is surprising that you do not have a research question or research goal. Moreover, the theory behind the hypotheses are lacking in the introduction. Why wouldn’t you expect a difference in O3I between rotating shift workers and day shift workers?  As it has been shown that shift workers consume poorer diets in general (10.1097/JOM.0000000000000088) and less healthy diets have been related to lower fish consumption?  Moreover, it is unclear why oyou look at intake of EPA +DHA below 0.5g and O3I>8%.

Methods

Could you please be more specific in what assessments were part of the normal occupational health check-up and what assessments were added for the current study.

Line 167-168: duration of fasting presumable fasting before blood was drawn, please specify this.

Line 186-887: How did you test whether the median O3I was below 8%? The mean is just a number, this does not need a statistical test? Why are you looking at DHA + EPA <0.5g?

Line 192-193: Please specify why these covariates were used in the analyses.

Line 194: between-group changes. What is ‘a group’ in your analyses?

Line 197: subject-specific random intercept. Do you mean participant-specific random intercept?  

Statistical assessment: the power calculation is missing, was your study powered for your primary outcome variable?

You mention a p of 0.05 however there is a vast number of analyses, a type II error seems very likely? It would be prudent to elaborate on this in the discussion.

General comment results: the large amount of numbers and information makes the text rather difficult to read. It can be questioned whether all information is necessary. For example the information how many people selected eggs and milk enriched with omega 3 at baseline and at follow-up. This numbers are so low, it might be more suitable for an appendix.

3.1.1. Study participants: it would probably be more clear to report these numbers in a table.

Figure 2: Would it prudent to remove the outlier of 405g/day, this does not seem to be a valid data point (2.8kg fish per week). Additionally, although it is interesting to see all the individual data points, it does make the figures (the same is true for figure 3 and 4) difficult to read. Possible these data points could be presented in a separate graph in a supplementary figure? I am also not sure why the data points are horizontally dispersed?

Table 1: why were age group 45-49 and smoker as reference group chosen? Additionally, why was age categorized as variable and not used as a continuous variable?

Table 2: what do the numbers below <4%, >4-<8 and >8% indicate?

Table 3: You executed a linear mixed model to look at the predicated within difference between baseline and follow-up. It is unclear to me what your predictor variable, outcome variable and level(s) are in this analyses? Why did you not simple execute an ANOVA or t-test for fish intake/DHA+EPA intake per day at baseline vs at follow-up? Additionally, it would be interesting to report the baseline values of the participants who did have follow-up data available especially as there were such distinct difference between those available for follow-up and those not.

Line 338-339: Please specify this with numbers; how often did they consume fish in the canteen.

Figure 5: This figure is very unclear. Additionally the exact same information is presented in text, please chose to either report in text or via figures.

Table 4: Were the analyses executed for each sociodemographic and occupation-related variable separate or were all variable entered in one model?

Line 406-408: Do not relate to previous sentence of average O3I?

Line 427-428: but the mean was still more than one percentage point lower than in the subset of VitaMinFemin participants.. does this point to male or female participants (I presume female, but please specify).

Line 437- 438 Occupational status has previous… How does this relate to the fact that manual and skilled workers had O3I lower than managerial staff?

Line 438-440: Background knowledge  in those employees… To whom does those employees refer?

Line 460: Vegans do not… There were no vegans in your study. Additionally, it is unclear why you mention pregnancy, your study was not focused on pregnancy?

Line 510-515: Unclear to which study this relates.

Line 532-533: through direct motivation.. did you use motivational intervention? In the methods you only describe an informational talk.

Line 538: 33000 employees, you only asked 1162 people to participate, how were these selected? This should be mentioned in the methods. Additionally the comparison of the study population to the overall population should be reported in the results not in the discussion.

Line 566: Only employees on day shifts could profit from the improved offer of fatty fish in the canteen. This could have caused the difference between managerial staff and manual workers. Managerial staff would have been able to consume their meal at the canteen every working day, while this is not the case for the manual workers who sometime work during the night when the canteen is not open. This is an important point which should be made more clear in you discussion.

Author Response

General comments: My major concern with the current study is the fact that it is not clear what the news value of the current study and that is unclear what the ‘intervention’ entailed exactly. There is a multitude of studies investigating the fish consumption and O3I of people from diverse population as you mention in the discussion; what does your study add? This should be specified in the introduction and at the beginning of the discussion. Additionally, you mention in the methods that people receive a consultation about health benefits of higher O3I and about the cut-offs, and that they received a handout. Additionally, you mention somewhere (rather briefly) that the amount of fish served in the canteen was increased. I feel if you want to discuss the follow up and ‘the effect of your intervention’ you should elaborate more on the intervention aspects. Especially as this is one of your hypotheses (hypothesis 3). In general there is so much information in the manuscript, that a specific focus is lacking.

Response: We thank the reviewer for the valuable comments. It is correct that several studies in the literature describe fish consumption and O3I from diverse populations. There are two unique features that we propose as the value of our study. First, we surveyed a working population in a land-locked part of Southern Germany where fish consumption is occasional rather than regularly and as recommended by national nutritional guidelines. Second, omega-3 fatty acid intake, O3I and clinical data from a large corporation is a truly rare dataset because of the strict ethical and employee protection laws in Germany that must be followed. Health data such as O3I are, however, essential to develop better strategies for improved employee health, wellness and productivity. In Section 4.4, we added the following text to address the reviewer’s concern: “The assessment of the omega-3 status in an occupational health environment is difficult to obtain due to ethical and employee protection regulations...”. The significance of the study with respect to employee health is now also discussed in this revised section.

Comment 1: Line 81-84: It is surprising that you do not have a research question or research goal. Moreover, the theory behind the hypotheses are lacking in the introduction. Why wouldn’t you expect a difference in O3I between rotating shift workers and day shift workers? As it has been shown that shift workers consume poorer diets in general (10.1097/JOM.0000000000000088) and less healthy diets have been related to lower fish consumption? Moreover, it is unclear why oyou look at intake of EPA +DHA below 0.5g and O3I>8%.

Response:

The study was set up as a monocentric baseline and follow-up survey without control group. We tested three hypotheses relating to omega-3 fatty acid intake, status and difference in subgroups of our study population. We appreciate the reviewer’s comment that the background of these hypotheses was not made sufficiently clear to the reader, and we accordingly added respective statements to the Introduction section to outline our hypotheses.

It is correct that the perceived health status has previously been described by Wirth et al. 2014 less favorable for shift workers compared to day workers when looking at specific dietary parameters. Nevertheless, there are fundamental differences between the study by Wirth et al. 2014 and ours. In our study we explored the relation of omega-3 fatty acid intake and erythrocyte concentrations rather than the quality of the diet in form of a diet inflammatory index score. And, from our point of view, it is important to note that a high intake of omega-3 fatty acids may not correlate per se with a high quality of the diet. Thus, the information gained form the NHANES analysis is of limited value when evaluating the O3I index in a work force health setting. Because data on omega-3 fatty acid intake, O3I and clinical health status from a large corporation is truly rare and this data is consequently very important for the development of additional health markers and eventually improved employee health strategies, we think the subject is worth to be studied.

We reason for an EPA + DHA intake recommendation of 0.5 g per day and added to the introduction the remark: “Intake recommendations for EPA and DHA vary widely, but range mostly between 0.2 and 0.5 g/d for healthy adults. For this study we recommend at the intake of at least 0.5 g EPA and DHA per day based on international expert opinions.” We refer to the recommendations by ISSFAL and the Academy of Nutrition and Dietetics. Moreover, we recommend a goal for O3I of above 8% based on the widely accepted recommendations described by Harris et al. (https://www.ncbi.nlm.nih.gov/pubmed/15208005) and von Schacky et al. (https://pubmed.ncbi.nlm.nih.gov/24566438/)

Comment 2: Could you please be more specific in what assessments were part of the normal occupational health check-up and what assessments were added for the current study.

Response: The normal occupational health check includes the following assessments: Anamnesis and medical consultation, clinical examination, laboratory diagnostic, urine diagnostic, fecal occult blood test and a questionnaire for self-examination of the personal health status. This information has been added to Section 2.1 of the revised manuscript.

Assessments added to the health check for our study were assessments of the O3I, lipoprotein (a) values and an omega-3 fatty acid specific questionnaire.

Comment 3: Line 167-168: duration of fasting presumable fasting before blood was drawn, please specify this.

Response: We appreciate this request for clarification and accordingly inserted “before blood was drawn” into the sentence.

Comment 4: Line 186-187: How did you test whether the median O3I was below 8%? The mean is just a number, this does not need a statistical test? Why are you looking at DHA + EPA <0.5g?

Response: We used simple one-sample one-sided median tests to formally check if median O3I and DHA+EPA values in our sample are significantly below our hypothesized values of respectively 8% and 0.5g respectively. We admit however that given baseline median values of 3.9% for O3I (with only 2 participants > 8%) and 221mg/d for EPA+DHA in our sample, do not require formal statistical testing. We deleted the corresponding passages in the methods and results accordingly.

Kindly refer also to our response to Comment 1 for a brief explanation why we were looking for EPA and DHA intakes lower than 0.5 g/day.

Comment 5: Line 192-193: Please specify why these covariates were used in the analyses.

Response: We selected these covariates for two reasons: Firstly, these variables might act as major confounders for the association between working time system and O3I. Secondly, besides the effect of shift work, we had a scientific interest in reporting the adjusted effects of the major sociodemographic/occupational-related variables on O3I as well. Due to the predominantly exploratory character of the study, these variables were not defined a priori, but rather judged as important confounders based on our past experiences. We tried to address this issue by providing more specific information in the Methods section of our revised manuscript which reads now as follows: “In order to test for significant differences in O3I levels between shift and day workers at baseline, we used a Mann-Whitney U test for the bivariate case and we applied a multiple linear regression model adjusted for age, occupational status, smoking, BMI, vegetarian diet, and eating in company restaurants, which we considered major potential confounders for the association between working time system and O3I”.

Comment 6: Line 194: between-group changes. What is ‘a group’ in your analyses?

Response: With the term group we meant the different categories of the sociodemographic, occupation- or lifestyle-related variables considered in our regression models (e.g. men, women, shift workers, <35y etc.). We now clarify what is meant with the term group, by changing the wording in the revised manuscript as follows: “In all models, we included the categorical group variable (e.g. categorical age, gender, BMI, etc.), time (baseline/follow-up) and group*time interaction terms as fixed effects […]

Comment 7: Line 197: subject-specific random intercept. Do you mean participant-specific random intercept?

Response: Yes, we mean participant-specific random intercept. We adapted our wording accordingly as follows: “[…] and a participant-specific random-intercept […]”.

Comment 8: Statistical assessment: the power calculation is missing, was your study powered for your primary outcome variable?

Response: This is an important comment of the reviewer and we accordingly added the following sentence in Section 2.1 to address this issue: “Due to the predominantly exploratory pilot character of our study, we did not carry out a formal power calculation for our primary outcome variable.”

Comment 9: You mention a p of 0.05 however there is a vast number of analyses, a type II error seems very likely? It would be prudent to elaborate on this in the discussion.

Response: The reviewer is correct that multiple testing is a serious issue. To address the reviewer’s concern, we added a sentence of caution to our Limitations section which reads as follows: “Finally, it should be noted that a variety of statistical tests were carried out, potentially increasing the possibility for a type I error (rejection of a true null hypothesis). Due to the predominantly exploratory character of our analyses we did however not formally adjust for multiple testing.

Comment 10: General comment results: the large amount of numbers and information makes the text rather difficult to read. It can be questioned whether all information is necessary. For example the information how many people selected eggs and milk enriched with omega 3 at baseline and at follow-up. This numbers are so low, it might be more suitable for an appendix.

Response: We agree with the reviewer’s point of view that our Results section contains a large amount of numbers and text. In order to improve readability of this section, we included a new Table 1 (please, see also our response to Comment 11), deleted the respective paragraph in Section 3.1.2 on milk and eggs consumption, shortened the text at the beginning of Section 3.1.4 as well as the text accompanying Figure 5.

Comment 11: 3.1.1. Study participants: it would probably be more clear to report these numbers in a table.

Response: We agree with the reviewer that an additional table might be very helpful in this regard and accordingly added the new Table 1 with information on the characteristics of the participants of the Omega-3 Employee Study at baseline with the company population on January 1st, 2019.

Comment 12: Figure 2: (a) Would it prudent to remove the outlier of 405g/day, this does not seem to be a valid data point (2.8kg fish per week). (b) Additionally, although it is interesting to see all the individual data points, it does make the figures (the same is true for figure 3 and 4) difficult to read. Possible these data points could be presented in a separate graph in a supplementary figure? I am also not sure why the data points are horizontally dispersed?

Response:

  1. We agree with the reviewer, that consuming 405 g lean fish per day is certainly unusual although not impossible. We re-evaluated the corresponding questionnaire and found the questions to be congruent to the calculated number. Apart from its magnitude, there is no hint for the data point to be invalid, we would therefore prefer to keep it in our dataset (although not in the figure, see next point).
  2. We agree that there is a tradeoff between readability and information contained in Figures 2 to 4. We tried to enhance readability by deleting the respective datapoints in Figure 2. Regarding Tables 3 and 4 we decided to keep the datapoints in the graph but reduced the opacity of the dots in order to improve readability.
  3. As noted by the reviewer, the datapoints in Figures 3 and 4 are horizontally dispersed. We added some random noise (“jitter”) to the data to see the amount of data more clearly. We tried to clarify things by adding the following sentence in the caption of Figures 2 to 4: “For illustrative purposes, some random noise (“jitter”) was added to the individual datapoints.”

Comment 13: Table 1: why were age group 45-49 and smoker as reference group chosen? Additionally, why was age categorized as variable and not used as a continuous variable?

Response: We decided to include age as a categorical variable as we noted some deviation from linearity in the association between age and O3I while visually inspecting a lowess plot in the upper part of the age distribution. We have therefore chosen the reference category of 40-44y (and not 45-49y) to specifically emphasize on the older and younger age group as compared to the middle ages. In order to keep the presentation of results consistent, we agree, however, that the choice of “smoker” as reference group might not have been well-thought and changed it accordingly to “nonsmoker” (respective changes were made also in Table 5).

Comment 14: Table 2: what do the numbers below <4%, >4-<8 and >8% indicate?

Response: In former Tables 1 and 2 (now Tables 2 and 3 of the revised manuscript), the numbers below <4%, >4-<8% and >8% indicated the (row) percentage of participants with least (<4%), average (4-8%) or largest (>8%) associated cardioprotection in the respective category. E.g., of those consuming any amount of lean fish, 49.2% fall into the category <4% [least cardioprotection], 50.5% into 4-8% [average cardioprotection] and 0.5% into >8% [largest cardioprotection]. In order to make things clearer, we added the variable labels of the categorical O3I (as defined in Section 2.5 “Variable Construction”) and also added %a in each column of the categorical O3I indicating that presented percentages are row percentages (changes made in both tables). For reasons of consistency, the same changes were applied to former Table 1 (now Table 2 of the revised manuscript).

Comment 15: Table 3: You executed a linear mixed model to look at the predicated within difference between baseline and follow-up. It is unclear to me what your predictor variable, outcome variable and level(s) are in this analyses? Why did you not simple execute an ANOVA or t-test for fish intake/DHA+EPA intake per day at baseline vs at follow-up? Additionally, it would be interesting to report the baseline values of the participants who did have follow-up data available especially as there were such distinct difference between those available for follow-up and those not.

Response: As proposed by the reviewer, we added the baseline values of the participants for which we have follow-up data to our revised Table 3 (now Table 4 in the revised manuscript). Regarding the mixed models, outcome variables were respectively fish intake/DHA+EPA simply regressed on time (baseline/follow-up) with a participant-specific random intercept. With “predicted” we wanted to figure out that results regarding within-change presented in the table are model-based estimations of the mean change from baseline to follow-up. As correctly noted be the reviewer, we could have also carried out a t-test for fish intake/DHA+EPA between baseline and follow-up which would be identical to our findings in the case of balanced data (no missing values). In fact, the results of both tests are quite similar between the mixed model and the paired t-test. The advantage of the mixed model is, however, that missing values between baseline and follow-up are better taken account of, since mixed models make use of all available observations (available case analysis) whereas t-tests applies listwise deletion (complete case analysis). We therefore decided to keep the statistical analysis here as is.

Comment 16: Line 338-339: Please specify this with numbers; how often did they consume fish in the canteen.

Response: We tried to specify this by adding the following sentence to the respective section: Of those participating at baseline and follow-up, respectively, 82, 27, and 181 participants stated to having increased, decreased or maintained their consumption of fatty fish in the company canteens (5 missing values were excluded).   

Comment 17: Figure 5: This figure is very unclear. Additionally the exact same information is presented in text, please chose to either report in text or via figures.

Response: We thank the reviewer for this very careful evaluation of our manuscript. For clarification, we added a legend to Figure 5 and extended the description in the caption of this figure: “Arrows indicate absolute and relative frequencies of a change in category from baseline to follow-up. E.g. 73 out of 147 participants (49.7%) whose baseline O3I was below 4% and who participated in baseline and follow-up, remained in the same category. On the contrary, 71 participants (48.3%) changed from <4% to ≥4-≤8%.“ In addition, to avoid duplication, we shortened the text referring to Figure 5.

Comment 18: Table 4: Were the analyses executed for each sociodemographic and occupation-related variable separate or were all variable entered in one model?

Response: The analyses were executed separately, as is now mentioned in the revised Methods section as follows: “We used linear mixed models to estimate within and between-group changes in O3I between baseline and follow-up for the main sociodemographic and occupational-related variables separately

Comment 19: Line 406-408: Do not relate to previous sentence of average O3I?

Response: The reviewer is correct. We changed the sentence to refer to the average O3I and the sentence reads now as follows: “In the Ludwigshafen Risk in Cardiovascular Health (LURIC) study investigators found higher average O3I values and O3I inversely correlating with mortality independent of other risk factors”

Comment 20: Line 427-428: but the mean was still more than one percentage point lower than in the subset of VitaMinFemin participants. does this point to male or female participants (I presume female, but please specify).

Response: This refers to the female participants of the VitaMinFemin study. “Female” was accordingly added to the sentence to clarify this for the reader.

Comment 21: Line 437- 438 Occupational status has previous… How does this relate to the fact that manual and skilled workers had O3I lower than managerial staff?

Response: The reviewer is correct and we are grateful for this helpful hint. The previous association of perceived occupational stress by Yong et al. does not refer to the association of O3I with occupational status in our current study. The sentence and reference were therefore deleted.

Comment 22: Line 438-440: Background knowledge in those employees… To whom does those employees refer?

Response: “Those” refers to the employees that chose to participate. The sentence was rephrased to improve understandability as follows: “Background knowledge on health and socioeconomic status in study participants…”

Comment 23: Line 460: Vegans do not… There were no vegans in your study. Additionally, it is unclear why you mention pregnancy, your study was not focused on pregnancy?

Response: It is correct that our study did not include vegan participants. The comparison to vegans is to emphasize that a previous study showed that even with a vegan lifestyle, with practically no dietary EPA and DHA intake, the O3I does not necessarily have to be as low as observed in our vegetarian participants not consuming fish. However, the beginning of the sentence was edited to take account for this surprising observation and reads now as follows: “Even vegans do not necessarily have lower O3I…”

The reviewer is right that the comparison to the situation of pregnancy may be misleading. Consequently, the two sentences were removed.

Comment 24: Line 510-515: Unclear to which study this relates.

Response: We appreciate this advice. The reference to our present study was added in the first sentence of the section. “Comparing baseline and follow-up in the present study…”

Comment 25: Line 532-533: through direct motivation.. did you use motivational intervention? In the methods you only describe an informational talk.

Response: Participants were provided respective information in form of a one-pager describing the potential health benefits of omega-3 fatty acids, a personal physician consultation and, when O3I results were available, an assessment of their O3I combined with dietary recommendations. As we agree with the reviewer, we accordingly replaced “direct motivation” with “information about the benefits” in the revised manuscript.

Comment 26: Line 538: 33000 employees, you only asked 1162 people to participate, how were these selected? This should be mentioned in the methods. Additionally the comparison of the study population to the overall population should be reported in the results not in the discussion.

Response: As described in Section 2.1, employees were “invited to also participate in the Omega-3 Employee Study” by the corporate health management within the voluntary occupational health check-up that is offered to employees once every three years. We agree with the reviewer’s point of view that the comparison of the study population to the overall population should be reported in the results. As already outlined in the response to “Comment 11”, we inserted a new Table 1, comparing baseline characteristics of the study participants with characteristics of the company population.

Comment 27: Line 566: Only employees on day shifts could profit from the improved offer of fatty fish in the canteen. This could have caused the difference between managerial staff and manual workers. Managerial staff would have been able to consume their meal at the canteen every working day, while this is not the case for the manual workers who sometime work during the night when the canteen is not open. This is an important point which should be made more clear in you discussion.

Response: This is a valid point raised by the reviewer. We addressed it by introducing the sentences “Employees on day shifts received an improved offer of fatty fish through the company-wide canteen system. This may have introduced bias against rotating shift workers who had less frequent access to the canteens.” to Section 4.5 of the revised manuscript.

Reviewer 3 Report

This study describes omega-3 status in 500 German workers. The manuscript is well written, but the study has some methodological limitations and I find the results to be of limited interest to the research community. The rather small number of participants from a selected group likely to be influenced by selection bias limits the interest in the first aim of the study, to describe mean omega-3 index. The second aim, to describe differences in O3I related to shift work is interesting. An additional question related to this aim is if there were differences in dietary intake between the groups? The effect of dietary advice in O3I effect of advice to is difficult to assess due to the lack of control group.  

Specific comments:

Methods:

Page 2, Line 88: A quasi-experimental design implies and intervention with a control group and does not correctly describe the design of this study.

Page 3, Lines 106-107 The rational for the cut-offs of O3I used in this study needs to be described and motivated. Reference 29 gives different potential cut-offs compared to reference 11. I lack a discussion regarding the choice in this study.

Page 4, Lines 130-132 Specify what you mean by “Most of the questions were adapted from the validated nutritional questionnaire” What part was not from a validated questionnaire?

Page 4, Lines 133-137 The methods used to analyze fatty acids need to be described better.

Page 4, Lines 147-149 This sentence does not make sense.

Page 4, Lines 184-185 and 192 A linear regression model is used, is the assumption of normality fulfilled? In line 184-185 a deviation from normality is stated.

Results:

Throughout the result section, many numbers are repeated in both text and tables/ figures.

Page 5, Lines 203-210 The section 3.1.1 is difficult to read.

Page 7, Line 259-260 The sentence with mean, sd, median, IQR and range is difficult to read and the results are better presented in such detail in the table.

Did the effect differ depending on baseline?

Discussion: The discussion gives a good description of other studies but is too long in general. It would have been interesting to include a section discussing genetic differences.

The conclusion to use O3I as a health variable is not supported by results of this study.

Author Response

General comments: This study describes omega-3 status in 500 German workers. The manuscript is well written, but the study has some methodological limitations and I find the results to be of limited interest to the research community. The rather small number of participants from a selected group likely to be influenced by selection bias limits the interest in the first aim of the study, to describe mean omega-3 index. The second aim, to describe differences in O3I related to shift work is interesting. An additional question related to this aim is if there were differences in dietary intake between the groups? The effect of dietary advice in O3I effect of advice to is difficult to assess due to the lack of control group.

Response: We appreciate the critical comments of the reviewer, but we do not fully agree with reviewer’s conclusion. We would like to highlight that our study contains data on omega-3 fatty acid rich food intake and an assessment of the O3I in population of relevant size, and it was conducted in the restricted environment of a large corporation with a representative population. We think the number of participants is large for this type of employee study. Study participation was entirely voluntary and naturally would favor those employees particularly interested in their own health and health effects of omega-3 fatty acids. The newly introduced Table 1 suggests, however, that the self-selected population of participants reflects the overall company population well. Moreover, despite the potential self-selection bias O3I values appeared rather low when comparing with previous studies in the literature.

Comment 1: Page 2, Line 88: A quasi-experimental design implies an intervention with a control group and does not correctly describe the design of this study.

Response: In order to make the design clearer, we changed our wording as follows: “For this monocentric baseline and follow-up survey without a control group […]

Comment 2: Page 3, Lines 106-107 The rational for the cut-offs of O3I used in this study needs to be described and motivated. Reference 29 gives different potential cut-offs compared to reference 11. I lack a discussion regarding the choice in this study.

Response: The reviewer correctly identified a mistake in the references referring to O3I cutoff values. O3I cutoff values for this study were chosen according to Harris et al. 2004. The rational for these cutoffs is that they are among of the earliest adapted and simplest for consumers or patients to understand. Moreover, for these cutoffs the largest body of evidence has amounted.

Comment 3: Page 4, Lines 130-132 Specify what you mean by “Most of the questions were adapted from the validated nutritional questionnaire” What part was not from a validated questionnaire?

Response: Questions which related specifically to our company (e.g. frequency of eating [fatty fish] in company restaurants), to omega-3 fatty acid supplementation or to the time fasted before participation are usually not part of validated questionnaires. We tried to clarify this issue in Section 2.3 which now reads as follows: “Several questions on omega-3 supplementation, time fasted before participation, and visiting company canteens were not obtained from the validated questionnaire.”

Comment 4: Page 4, Lines 133-137 The methods used to analyze fatty acids need to be described better.

Response: A description of the gas chromatography method was added to Section 2.4.

Comment 5: Page 4, Lines 147-149 This sentence does not make sense.

Response: We thank the reviewer for the careful reading of our manuscript, and we accordingly rephrased the sentence as follows: Lipoprotein (a) values were classified into three categories, "no indication of Lp(a) associated atherogenic risk" (≤30 mg/dL), "Lp(a) associated risk slightly increased" (>30-≤60 mg/dL) and "Lp(a) associated risk significantly increased" (> 60 mg/dL).

Comment 6: Page 4, Lines 184-185 and 192 A linear regression model is used, is the assumption of normality fulfilled? In line 184-185 a deviation from normality is stated.

Response: Visual inspection of the residuals via normal-quantile-quantile-plot after estimating our multivariable model showed some slight deviation from normality, potentially affecting estimated p-values. Thus, log-transformation of our outcome variable was considered. However, since interpretation of estimated coefficients in a log-linear model is not straight-forward, and since all of the categories/variables were consistently significant or insignificant in both models (except for the category “several times per week” of variable “eating in company restaurants”, which was significant in the log-linear model only), we decided to stick with the linear model for ease of interpretation.

Comment 7: Throughout the result section, many numbers are repeated in both text and tables/ figures.

Response: The reviewer is correct. We tried to reduce redundancy in the revised manuscript in Sections 3.1.1., 3.1.4. and 3.2.3 wherever possible.

Comment 8: Page 5, Lines 203-210 The section 3.1.1 is difficult to read.

Response: We admit that the readability of the respective section could be improved. In order to address the reviewer’s concern and also Comment 11 of Reviewer #2, we inserted a new Table 1 and, accordingly, shortened the text.

Comment 9: Page 7, Line 259-260 The sentence with mean, sd, median, IQR and range is difficult to read and the results are better presented in such detail in the table.

Response: We agree with the reviewer’s opinion that the corresponding sentence might be difficult to read. The data described in this sentence is already visible in higher detail in Figure 3 and Table 2. We shortened the text and referred to the aforementioned figure and table.

Comment 10, Reviewer #3: Did the effect differ depending on baseline?

Response: We apologize as we are not entirely sure as to which effect the reviewer refers. We used a mixed model approach to answer the question whether the mean change in O3I from baseline to follow-up differed between the respective groups (the difference in differences from baseline to follow-up), which is directly captured via the time/group interaction term(s). Thus, the model contained baseline and follow-up O3I values as an outcome, entering baseline values as additional covariates would not have been feasible in this regard. We admit however, that we cannot exclude the possibility, that regression towards the mean might have occurred and acknowledged this in our Limitations section as follows: “A further limitation which should be acknowledged it the possibility for regression towards the mean, meaning that outlier observations at baseline tend to be followed by values that are closer to the average values at follow-up.

Comment 11: Discussion: The discussion gives a good description of other studies but is too long in general. It would have been interesting to include a section discussing genetic differences.

Response: We agree that the discussion is lengthy and shortened Section 4.2. The genetic differences with respect to endogenous synthesis or absorption efficacy of EPA and DHA were briefly referred to in the second paragraph of the introduction. In order to stay focused we decided not to further elaborate on the topic in our Discussion section.

Comment 12: The conclusion to use O3I as a health variable is not supported by results of this study.

Response: The reviewer is correct. Our intend for this study was to test whether the determination of O3I was useful to survey the baseline omega-3 fatty acid status in an employee population and to monitor whether changes can be achieved by a consultative/informative intervention. We consider this an important information for the further development of O3I into a potentially useful health variable and for developing new public health strategies for improving dietary intake of EPA and DHA.

Round 2

Reviewer 2 Report

I thank the authors for the thorough and elaborate answers to my previous comments. They have suffiently  answered most questions and I feel the manuscript has improved accordingly. I am however still not completely sure what the ‘news value’ is, I am not sure whether I feel that the fact that this study was executed in an occupational health environment (i.e., a company) does not give it news value above previous population based studies. Additionally I have some minor comments:

Line 55-56: For this study we recommend at the intake of at least 0.5 g EPA and DHA per day based on international expert opinions [11, 12].

Should this be recommended?

Line 296: through supplements to average half a capsule or 0.5 g oil containing 210 mg EPA plus DHA [35]. This increased the average EPA plus DHA intake at baseline by 15 mg to 221 mg/d,

I do not understand how the consumption of 210mg of EPA + DHA per day in capsule form led to an increase of 15mg EPA plus DHA?

Line 324: The average O3I for all participants at baseline (n=500) was 4.1% (SD: 1.1%) and 3.9% (IQR: 3.3-4.7%) for the mean and median O3I

Sentence is not correct, the average cannot be both the mean and the median.

Figure 3: “For illustrative purposes, some random noise (“jitter”) was added to the individual datapoints.”

Maybe add ‘this add horizontal dispersion’ to make it even more clear.

Line 347: The small group of participants indicating vegetarian lifestyle and not to consume fish contributed the lowest subcategory in this study (median 2.7%, Table 1).  

Should this be table 2. Additionally should you specify median O3I?

Line 469 Regular use of the company canteen was another predictor for higher O3I mean and median values, as was the choice of fatty fish on offer in the canteen (Table 1).

Should this be table 3?

Figure 5: Can you add the percentage of people that move to the ‘green’ category?

General: I am missing a description of how the seafood offer in the canteen was improved.

Line 670: However, vegetarian not consuming fish had very low median O3I of 2.7%. Even vegans do not necessarily have lower O3I compared to omnivorous subjects if the latter consume very little omega-3 fatty acids [43].

These two sentences seem to contradict each other. Vegans do no necessarily have a lower O3I than omnivores, but still you found a lower median O3I in strict vegetarians in your sample?

Line 702: resulted in a median or mean O3I

Mean or median? Shouldn’t this be one of the two?

Line 754: There was, however, a slightly lower relative study participation of men, manual workers, and rotating shift employees. This might indicate a lower health interest in these subgroups.

Was this difference in participation significant. Additionally, do you expect that this influences the results of your study?

Line 766: the distribution of study participants compared reasonably well with the employee demographics at the company’s Ludwigshafen site data from 2018 with respect to age, gender, occupational status, work shift system and health related parameters such as BMI and smoking status.

In your table there is no information with regard to health related parameters for non-participants. The sentence above does suggest that this is available. Could you clarify?

Line 789: The ethical aspects relate to asking participants for an additional blood draw (follow-up), in a study of which no apparent benefits would be available to those assigned to the control group

Sentence is not grammatically correct.

Author Response

General comment: I thank the authors for the thorough and elaborate answers to my previous comments. They have suffiently answered most questions and I feel the manuscript has improved accordingly. I am however still not completely sure what the ‘news value’ is, I am not sure whether I feel that the fact that this study was executed in an occupational health environment (i.e., a company) does not give it news value above previous population based studies. Additionally I have some minor comments:

Response: We thank the reviewer for the valuable comments. We feel that the study group offers a rare opportunity to investigate in employees the omega-3 fatty acid intake, relate this to the O3I and observe the extent to which the O3I can be modified through omega-3 fatty acid intake. Our data will help the further assessment and development of health markers for more comprehensive occupational health support. Moreover we further shortened the text and improved clarity of the study.

Comment 1: Line 55-56: For this study we recommend at the intake of at least 0.5 g EPA and DHA per day based on international expert opinions [11, 12].

Should this be recommended?

Response: We appreciate the thorough review. This mistake was corrected: “For this study we recommended the intake of…”

Comment 2: Line 296: through supplements to average half a capsule or 0.5 g oil containing 210 mg EPA plus DHA [35]. This increased the average EPA plus DHA intake at baseline by 15 mg to 221 mg/d,

I do not understand how the consumption of 210mg of EPA + DHA per day in capsule form led to an increase of 15mg EPA plus DHA?

Response: The reviewer is correct in that we should have provided a better explanation of how we approximated the EPA plus DHA intake for the entire study population and a reference to Table 2. Previous Table 4 was thus moved up and is now assigned Table 2, consequently tables were renumbered (for technical reasons the move of the table move could not be tracked).

The reference to the calculation was now clarified by revising the text as follows: “through supplements to average half a capsule or 0.5 g oil containing 210 mg EPA plus DHA [35]. Adding this additional omega-3 fatty acid intake fraction to the intake of the total baseline population increased the average EPA plus DHA intake at baseline by 15 mg to 221 mg/d (Table 2),”

Comment 3: Line 324: The average O3I for all participants at baseline (n=500) was 4.1% (SD: 1.1%) and 3.9% (IQR: 3.3-4.7%) for the mean and median O3I.

Sentence is not correct, the average cannot be both the mean and the median.

Response: The reviewer is correct. We revised the sentence to: “The mean and median O3I at baseline (n=500) were 4.1% (SD: 1.1%) and 3.9% (IQR: 3.3-4.7%), respectively.”.

Comment 4: Line Figure 3: “For illustrative purposes, some random noise (“jitter”) was added to the individual datapoints.” Maybe add ‘this add horizontal dispersion’ to make it even more clear.

Response: To make the legend in Fiugres 3 and 4 more clear the sentence now states: “For illustrative purposes, horizonal dispersion (“jitter”) was added to the individual datapoints”

Comment 5: Line 347: The small group of participants indicating vegetarian lifestyle and not to consume fish contributed the lowest subcategory in this study (median 2.7%, Table 1).

Should this be Table 3. Additionally should you specify median O3I?

Response: We appreciate the reviewer catching these oversights and corrected the Table number in the renumbered table order and specified “O3I” in the text.

Comment 6: Line 469 Regular use of the company canteen was another predictor for higher O3I mean and median values, as was the choice of fatty fish on offer in the canteen (Table 1).

Should this be Table 3?

Response: We appreciate the reviewer catching this mistake which we corrected in the renumbered order of Tables.

Comment 7: Figure 5: Can you add the percentage of people that move to the ‘green’ category?

General: I am missing a description of how the seafood offer in the canteen was improved.

Response: As suggested by the reviewer we added the percentage of people that move to the ‘green’ category. We also added a sentence describing the improved seafood offer in the canteen in the second paragraph of section 2.1.: “Participants were recommended to take advantage of the improved offer of fatty seafood in the canteen consisting of two additional fatty fish meals per week.”.

Comment 8: Line 670: However, vegetarian not consuming fish had very low median O3I of 2.7%. Even vegans do not necessarily have lower O3I compared to omnivorous subjects if the latter consume very little omega-3 fatty acids [43].

These two sentences seem to contradict each other. Vegans do no necessarily have a lower O3I than omnivores, but still you found a lower median O3I in strict vegetarians in your sample?

Response: The authors appreciate the advice that this section is not clear. We wanted to highlight that in our study a vegetarian diet containing milk and eggs resulted in lower O3I values than previously described for a vegan diet only based on plant products. We rephrased the sentences and hope to be clearer: “In our study vegetarians not consuming fish had a very low median O3I of 2.7%. This seemed unexpectedly low, because vegans do not necessarily have lower O3I than omnivorous subjects, both consuming very little omega-3 fatty acids [43].”

Comment 9: Line 702: resulted in a median or mean O3I

Mean or median? Shouldn’t this be one of the two?

Response: We appreciate this request for clarification. Both, median and mean resulted in the same value. We used “and” instead of “or”: “…resulted in a median and mean O3I increase of 0.5 percentage points…”

Comment 10: Line 754: There was, however, a slightly lower relative study participation of men, manual workers, and rotating shift employees. This might indicate a lower health interest in these subgroups.

Was this difference in participation significant. Additionally, do you expect that this influences the results of your study?

Response: We appreciate this request for clarification. Using Pearson’s Chi-squared test, total participants in our study differed significantly from the total company population regarding gender (p = 0.004), occupational status (p<0.001) and working time system (p<0.001. There was no significant difference regarding categorical age. We added this finding in table 1 and updated our limitations section as follows: “There was, however, a slightly lower relative study participation of men, manual workers, and rotating shift employees (Table 1). This might indicate a lower health interest in these subgroups and could have affected study results in favor of an over-estimation of the effect of our intervention.”

Comment 11: Line 766: the distribution of study participants compared reasonably well with the employee demographics at the company’s Ludwigshafen site data from 2018 with respect to age, gender, occupational status, work shift system and health related parameters such as BMI and smoking status.

In your table there is no information with regard to health related parameters for non-participants. The sentence above does suggest that this is available. Could you clarify?

Response: The reviewer is right. We oversaw that the data on health-related parameters was currently not available for the company population. Consequently, we removed these parameters from the text.

Comment 12: Line 789: The ethical aspects relate to asking participants for an additional blood draw (follow-up), in a study of which no apparent benefits would be available to those assigned to the control group

Sentence is not grammatically correct.

Response: We appreciate this request for clarification. The sentence now reads: “From an ethical perspective it was not reasonable to include a control group. Control participants would have been asked for an additional blood draw (follow-up) without apparent study benefits.”

Reviewer 3 Report

Dear Madam/ Sir,

Thank you for providing an updated version of this manuscript. I find that the manuscript has improved significantly and is easier to read. I still find the study to be of limited significance and interest.

Best,

Author Response

General comments: Thank you for providing an updated version of this manuscript. I find that the manuscript has improved significantly and is easier to read. I still find the study to be of limited significance and interest.

Response: We appreciate the critical comments of the reviewer. We still feel that the study group offers a rare opportunity to investigate in employees the omega-3 fatty acid intake, relate this to the O3I and observe the extent to which the O3I can be modified through omega-3 fatty acid intake. Our data will help the further assessment and development of health markers for more comprehensive employee health support.